



# Effect of changing ocean circulation on deep ocean temperature in the last millennium

Jeemijn Scheen[1,2] and Thomas F. Stocker[1,2]

[1]Climate and Environmental Physics, Physics Institute, University of Bern, Bern, Switzerland
[2]Oeschger Centre for Climate Change Research, University of Bern, Bern, Switzerland

**Correspondence:** Jeemijn Scheen (jeemijn.scheen@climate.unibe.ch)

**Abstract.** Paleoreconstructions and modern observations provide us with anomalies of surface temperature over the past millennium. The history of deep ocean temperatures is much less well-known and was simulated in a recent study for the past 2000 years under forced surface temperature anomalies. In this study, we simulate the past 800 years with an illustrative forcing scenario in the Bern3D ocean model, which enables us to assess the role of changes in ocean circulation on deep ocean temperature. We quantify the effect of changing ocean circulation by comparing transient simulations (where the ocean dynamically adjusts to anomalies in surface temperature – hence density) to simulations with fixed ocean circulation. We decompose temperature, ocean heat content and meridional heat transport into the contributions from changing ocean circulation and changing sea surface temperature (SST). In the deep ocean, the contribution from changing ocean circulation is found to be as important as the changing SST signal itself. Firstly, the small changes in ocean circulation amplify the Little Ice Age signal around 3 km depth by at least a factor of two, depending on the basin. Secondly, they fasten the arrival of this atmospheric signal in the Pacific and Southern Ocean at all depths, whereas they delay the arrival in the Atlantic between about 2.5 and 3.5 km by two centuries. This delay is explained by an initial competition between the Little Ice Age cooling and a warming due to an increase in relatively warmer North Atlantic Deep Water at the cost of Antarctic Bottom Water. Under the consecutive AMOC slowdown, this shift in water masses is inverted and aging of the water causes a late additional cooling. Our results suggest that small changes in ocean circulation can have a large impact on the amplitude and timing of ocean temperature anomalies below 2 km depth.

## 1 Introduction

The climate period from 1200 to 1750 CE manifests modes of natural variability and response to solar and volcanic forcing without the substantial anthropogenic radiative forcing from increased greenhouse gas concentrations (Masson-Delmotte et al., 2013). A growing paleoclimatic database of surface reconstructions exists that quantifies this natural variability with at least continental resolution (Mann et al. (1998, 2009); McGregor et al. (2015); Neukom et al. (2019); PAGES 2k Consortium: Ahmed et al. (2013) and Emile-Geay et al. (2017)). Early instrumental records, based on surface measurements and their combination with models, permit the separation of climate responses due to solar and volcanic forcing respectively (Brönnimann et al., 2019), but oceanic reconstructions below the sea surface are much scarcer, especially in the deep ocean (Abraham et al., 2013;



Levin et al., 2019). For example, only 17 % of the ocean cores used in McGregor et al. (2015) lie below 2000 m and only 2 % below 3000 m. Therefore, models can be of great value by simulating past and present deep ocean temperatures in agreement with paleoclimatic reconstructions at the surface.

This study focuses on the propagation of atmospheric temperature anomalies into the deep ocean and is motivated by Gebbie and Huybers (2019), who investigated this during the past 2000 years. They used an ocean model with fixed circulation, which

they inferred from modern observations and constrained by measurements of the legendary HMS *Challenger* expedition of 1872-1876. The changes they found in deep ocean temperature in the Atlantic and Pacific are delayed responses to variations of the forced surface climate. We wonder to what extent their results would be different with a model that exhibits dynamically changing circulation. In this study, we test this with the Bern3D model.

Previous studies have pointed out that temperature cannot always be approximated as a passive tracer, since it induces

circulation changes, which influence the patterns of heat uptake (Banks and Gregory, 2006; Xie and Vallis, 2011; Winton et al., 2013; Marshall et al., 2015; Garuba and Klinger, 2016). These authors used various approaches to separate the effects of changing circulation (also called redistribution transport) and changing sea surface temperature (SST). Banks and Gregory (2006), Marshall et al. (2015) and Garuba and Klinger (2016) used a passive tracer that diagnoses the effect of changing SST only, whereas Xie and Vallis (2011) used a tracer diagnosing the effect of changing circulation only, and Winton et al. (2013)

ran simulations with artificially fixed (FIX) ocean circulation in addition to transient (TRA) simulations with dynamically changing circulation. Here we follow the latter approach, although with ocean-only simulations in order to make sure that fixed and transient simulations undergo the exact same surface boundary conditions. By prescribing these SST and sea surface salinity (SSS) fields, we cannot quantify differences in global ocean heat uptake anymore, as was done in Winton et al. (2013). In exchange, this allows us to quantify the downward propagation of small temperature anomalies to the abyss and the spatial

pattern of heat uptake without biases due to ocean-atmosphere feedbacks, which differ between transient and fixed simulations.

The paper is organized as follows. Section 2 presents the model and the methods needed to disentangle surface signal changes from those caused by circulation changes. In Sect. 3, we analyse the propagation of temperature signals into the deep ocean and decompose them according to different physical mechanisms. Section 4 investigates the causes of leads and lags registered at depth in the different ocean basins, including their sensitivity on changes in mixing and wind stress. In Sect. 5, we critically

compare our results with the study of Gebbie and Huybers (2019) and we conclude in Sect. 6.

## 2    Modelling framework

### 2.1    Model description

We use an Earth system Model of Intermediate Complexity (EMIC), the Bern3D model version 2.0, developed by Ritz et al. (2011) and systematically constrained by observations (Roth et al., 2014). The model consists of a 3-dimensional dynamic

ocean circulation model with 32 depth layers coupled to a 1-layer atmosphere and a land biosphere. The ocean component of the model is based on frictional geostrophic balance equations (Edwards et al., 1998; Müller et al., 2006), and the atmosphere consists of an energy balance model including a parametrized hydrological cycle (Ritz et al., 2011). The Bern3D has an





**Table 1.** Simulations performed in this study.

| Simulation | Model[a] | Ocean[b] | Parameters[c] | Notes |
| --- | --- | --- | --- | --- |
| OcAtmTRA | coupled | transient | - | generates SST/SSS/sea ice area fields each time step |
| OcTRA | ocean | transient | - | standard simulation |
| OcFIX | ocean | fixed | - | standard simulation |
| OcTRA_weakmix | ocean | transient | $K_D \times \frac{1}{2}$ | |
| OcFIX_weakmix | ocean | fixed | $K_D \times \frac{1}{2}$ | |
| OcTRA_strongmix | ocean | transient | $K_D \times 2$ | |
| OcFIX_strongmix | ocean | fixed | $K_D \times 2$ | |
| OcTRA_weakwind | ocean | transient | $\tau_{wind} \times \frac{1}{2}$ | |
| OcFIX_weakwind | ocean | fixed | $\tau_{wind} \times \frac{1}{2}$ | |
| OcTRA_strongwind | ocean | transient | $\tau_{wind} \times 2$ | |
| OcFIX_strongwind | ocean | fixed | $\tau_{wind} \times 2$ | |

[a] Simulations are either run using the coupled Bern3D (with radiative forcing in the atmosphere as in Fig. 1a) or as ocean-only Bern3D (prescribed SST, SSS and sea ice from OcAtmTRA).

[b] Ocean circulation can be fixed or transient, i.e., responding to SST and SSS changes.

[c] In sensitivity simulations, the mixing parameter $K_D$ (diapycnal diffusivity) and wind stress $\tau_{wind}$ are varied.

equilibrium climate sensitivity of $3.0°\text{C}$ in the standard version. Seasonally varying winds are added as a fixed forcing. Sea-ice growth and melt are thermodynamically simulated. The horizontal grid resolution is $40 \times 41$ cells, and the time step is 3.8 days
(Roth et al. (2014), Appendix A). Because of its relatively coarse resolution and reduced complexity in the atmosphere, the model is well suited for long simulations and sensitivity studies. For example, 5,000 model years can be simulated within 24 hours.

Here, we use the idealized age tracer, which provides information about changes in the absolute strength of the circulation, and a number of dye tracers to identify relative changes in water masses. The idealized ocean age tracer (England, 1995; Hall
and Haine, 2002) records the time since water was last in contact with the atmosphere. It satisfies the transport equation like any other conservative tracer implemented in the ocean model with the additional production term $1 \, \text{yr} \, \text{yr}^{-1}$. When water reaches the surface, the ideal age is reset to zero. Dye tracers track the water that originates from a certain surface area of the ocean, e.g., the Southern Ocean (Fig. A1). At the surface we restore the tracer concentration to the value 100 % in the area of origin of the specific dye tracer, and 0 % elsewhere. Restoring is not instantaneous, but occurs on a time scale of 20 days in
order to avoid numerical problems. Ideal age and dye tracers are spun up simultaneously with the circulation such that they are at their equilibrium distribution at the start of the experiments.



## 2.2 Radiative forcing of medieval cooling and early industrial warming

We simulate the past 800 years followed by 800 years in the future, using an idealized radiative forcing (Fig. 1a). Simulations start from a pre-industrial steady state at 1200 CE. The amplitudes of the medieval cooling (1200-1750 CE), also referred to as
Little Ice Age (LIA), and the subsequent industrial warming (1750-2000 CE) are followed by 800 years of constant radiative forcing. Our illustrative scenario represents this by a linear decrease in radiative forcing by $-0.55 \, \mathrm{Wm^{-2}}$ over 1200-1750 CE followed by an increase to $+1.4 \, \mathrm{Wm^{-2}}$ from 1750 to 2000 and constant thereafter. Possible radiative forcings due to land-use change, volcanic eruptions or solar irradiation are not considered in our illustrative scenario.

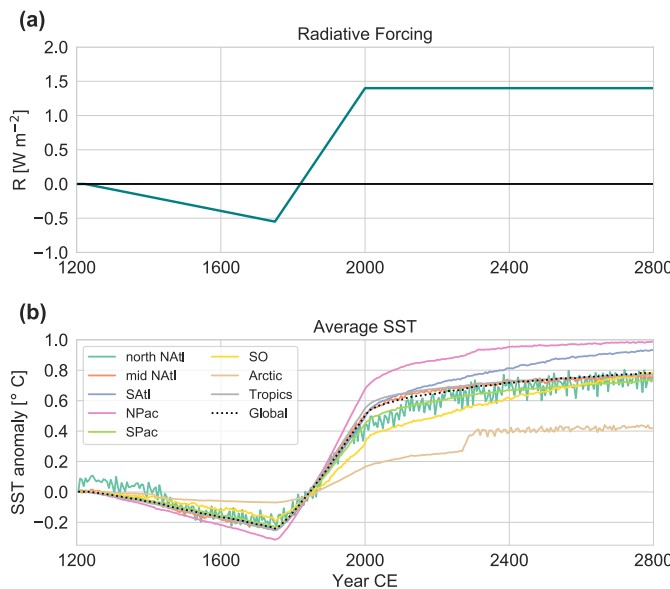

**Figure 1. (a)** Idealized radiative forcing, which is used in the case of coupled simulations (OcAtmTRA and OcAtmFIX, see Table 1). **(b)** Sea surface temperature anomaly as a result of the forcing in (a): globally (black, dotted line) and averaged per region (colors, defined in Fig. A1). The global average SST at 1200 CE is $18.0°$C.

While the exact radiative forcing is of lesser relevance here, we note that the global average SST anomalies are comparable
with reconstructions (Ahmed et al., 2013; Rayner et al., 2003). Our global SST anomaly shows a cooling amplitude of $-0.24$ K followed by a $+0.74$ K warming until 2000 CE (Fig. 1b) and approximately follows the SST variations reported by Gebbie and Huybers (2019) ($-0.27$ K; $+0.87$ K; from their Fig. 1a), which are in turn based on proxy reconstructions from the PAGES 2k Consortium (Ahmed et al., 2013) before 1950 CE and instrumental data from HadISST after 1870 CE (Rayner et al., 2003). For the overlap period (1870-1950), Gebbie and Huybers (2019) take a weighted linear combination of the two such that their
resulting warming amplitude ($+0.87$ K) is higher than the proxy-only signal (Ahmed et al. (2013): $+0.53$K), because the instrumental data exhibit a larger amplitude of industrial warming.





## 2.3   Simulations with fixed and transient ocean circulation

Quantification of the influence of changing ocean circulation on the tracer distributions of temperature and salinity is not trivial, because temperature and salinity are 'active' tracers, which influence the ocean circulation through changes in density.

We disentangle the effects caused by changes in the surface values from those caused by circulation changes by requiring for every experiment two simulations: one in which the ocean circulation adjusts transiently to the buoyancy distribution and one in which the circulation is kept constant, both forced with identical surface boundary conditions.

Simulations with fixed ocean circulation are implemented by using diagnosed values at every grid cell from one annual cycle at steady state. Therefore, the circulation, including advection, diffusion and convection, is maintaining a perpetual seasonal

cycle and does not respond to SST changes impacted by atmospheric changes at the ocean surface. Note that unphysical effects could arise, e.g., when convection still operates, although the water column at that location could now be stably stratified. This approach allows us to isolate the effect of changing ocean circulation by taking the difference between a transient and a fixed run under the same boundary conditions. This is realized by a series of ocean-only simulation pairs, summarized in Table 1.

Crucial to our approach is that these boundary conditions must be identical at the atmosphere-ocean interface and not,

e.g., at the top of the atmosphere. When we use the fully coupled Bern3D with transient ocean circulation (OcAtmTRA) and with fixed circulation (OcAtmFIX; not used elsewhere) under the same radiative forcing, the resulting two simulations possess different SST and SSS patterns, because ocean-atmosphere feedbacks differ between the transient and fixed case, as was already suggested by Garuba and Klinger (2016). These SST differences are small, but of the same order of magnitude as the changes in deep ocean temperature, and would therefore mask the signal we want to detect. We resolve this by running the model in an

ocean-only mode that is forced with the same SST, SSS and sea ice in both transient and fixed simulations. The possibility to run the Bern3D model under time-varying surface boundary conditions has been newly implemented for this study. For every time step of the 1600 simulation years, the SST, SSS and sea ice at every surface grid cell are set to a latitude-longitude field that is obtained from a previous coupled transient simulation OcAtmTRA over 1600 years (Fig. 1). All ocean-only simulations in Table 1 are run under identical surface boundary conditions obtained from the OcAtmTRA simulation from year 1200 to

2800 CE.

## 2.4   Steady state

The coupled simulations are run to equilibrium under pre-industrial conditions. When switching to the ocean-only configuration, a model drift is observed that requires an extension of the spin-up. The ocean-only model is run for another 5000 years using the steady state SST, SSS and sea ice distributions from a period of 100 years, repeated 50 times. Afterwards, the ocean-

only simulations OcTRA and OcFIX are started. During the extended spin-up, adjustments of the global ocean circulation take place: the steady state Atlantic meridional overturning circulation (AMOC) maximum decreases by 1.5 Sv and the strength of the global meridional overturning circulation (MOC) decreases by 0.8 Sv. Deep ocean temperatures are 0.3 to 0.6°C colder below 3 km, because of a relative increase of Antarctic Bottom Water (AABW) with respect to North Atlantic Deep Water (NADW), and are up to 0.2°C warmer in the depth range from 500 to 3000 m due to an absolute decrease in NADW. By the





end of this additional spin-up, the residual drift of basin-mean ocean temperature at 3 km depth during the last 500 simulation
years is less than $4 \cdot 10^{-6}$ K/yr in the Pacific and less than $2 \cdot 10^{-6}$ K/yr in the Atlantic. Simulations with varying mixing and
wind stress require their own steady state. For each of these four simulation pairs, we perform a similar 5000 year ocean-only
spin-up, but with the respective parameter values given in Table 1.

## 3 Results and discussion

### 3.1 Circulation change


Figure 2 shows the responses of the MOCs in the Atlantic (AMOC), the Indo-Pacific (IPMOC) and Southern Ocean (SOMOC)
for the ocean-only simulations forced by the imposed surface fields during the 1600 simulation years. Maxima and minima
are evaluated below a depth of 400 m in order to avoid shallow Ekman-driven overturning cells. The Atlantic, Pacific and
Indian basins are defined between 35°S and up to 70°N, and the Southern Ocean (SO) extends southward of 35°S throughout
this study, unless indicated otherwise. By design, the circulation is constant over time in the fixed case. During the medieval
cooling, the AMOC slightly increases by ∼0.2 Sv, whereas the negative IPMOC and SOMOC both strengthen by ∼0.7 Sv.
Industrial warming starting in 1750 causes the AMOC to slow down by 1.5 Sv, whereas both the IPMOC and SOMOC weaken
by about ∼2.2 Sv until 2000. This 1.5 Sv AMOC slowdown is within the $1\sigma$ CMIP5 range of fully coupled AOGCMs and the
AMOC in that model mean experiences a similar decrease of 2 Sv from 2000 to 2100 CE under low emission forcing (Collins
et al., 2019). After 2000, the AMOC maximum recovers under the constant forcing with a relaxation time scale of about 470
yr, whereas the IPMOC takes significantly longer to reach equilibrium, and for this case a well-constrained time scale cannot
be determined from the present simulations. The SOMOC does not recover, but weakens by an additional ∼1.3 Sv over the
final 800 years.

The overturning stream functions are shown in Fig. 3 at 1205 CE and their anomalies at years 1750 CE and 2000 CE
(simulation OcTRA). The AMOC deepens during the medieval cooling phase, which leads to an increase of about 0.8 Sv at
2 km depth, although the maximum AMOC increased only by ∼0.2 Sv by 1750 (Fig. 1a). The largest changes of the AMOC
occur during the industrial warming at depths of 2 km reaching up to 2 Sv, and the reduction of overturning occupies the
entire Atlantic ocean basin from 1 to 3 km depth. IPMOC changes are primarily located in the southern hemisphere with a
strengthening of the circulation below 3 km at the southern margin of the basin during the medieval cooling, and a subsequent
weakening above 3 km. The global MOC shows a strong bipolar response to the medieval cooling and industrial warming.
That is, the absolute value of MOC strength increases in both hemispheres from 1205 to 1750, and a subsequent opposite and
stronger response is simulated during the industrial warming.

### 3.2 Propagation of temperature anomalies into the deep ocean

We now turn to the propagation of the atmospheric temperature anomaly into the deep ocean. Figure 4a, b show Hovmöller
diagrams illustrating the development of the basin-averaged temperature anomalies as a function of depth for each ocean basin.



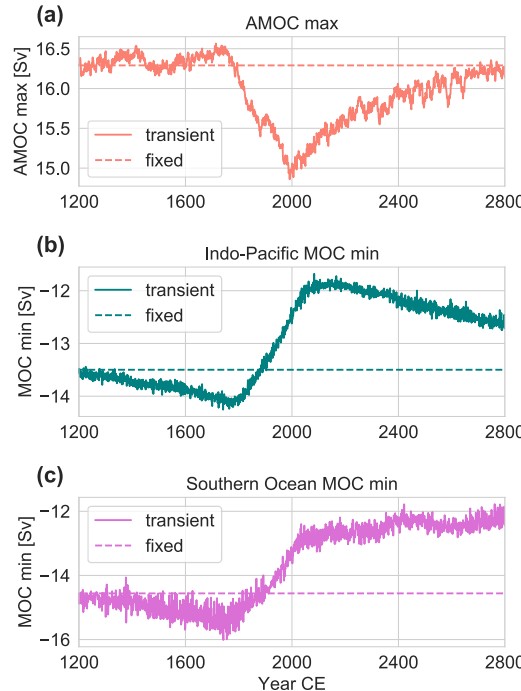

**Figure 2.** Response of the maximum respectively minimum values of the meridional overturning circulation (MOC) to imposed SST, SSS and sea ice (Fig. 1b). **(a)** Atlantic MOC maximum, **(b)** minimum of the MOC in the combined Pacific and Indian basins (IPMOC) and **(c)** in the Southern Ocean (SOMOC). Simulations OcTRA (transient) and OcFIX (fixed) are shown.

Temperature anomalies are small and reported here in units of centi-Kelvin (1 cK = $10^{-2}$ K). The propagation of the LIA cooling into the deep ocean is clearly visible, followed by the industrial warming signal. Qualitatively distinct patterns for the transient (TRA) and fixed (FIX) circulations emerge in all basins with a generally faster and stronger transfer of the anomalies to the deep ocean in the TRA case. However, local differences appear, particularly in the Atlantic Ocean. For TRA, the deep
Atlantic shows an isolated cold anomaly at about 3 km depth, which develops by the end of the medieval cooling period and persists for several centuries. This feature, observed in the more realistic TRA simulation, suggests that the cold anomaly created by the pre-industrial cooling can be detected long into the future. It is evident that this anomaly is a result of the changes in MOC, as the fixed circulation case does not exhibit this isolated anomaly. At year 2000 in TRA, the cooling is still strong and occurs below 2.5 km depth in the Atlantic, whereas for FIX much weaker cooling is confined to below 3 km. On the other
hand, in the Pacific basin, where the MOC changes during the warming are generally smaller, the signal propagation into depth is rather uniform and follows the surface forcing with a delay due to ocean circulation. Here, the faster signal transfer to depth in TRA is particularly evident. The SO also shows a uniform signal propagation with a faster transfer in TRA.

In the Atlantic, the more effective downward propagation of negative temperature anomalies in TRA is linked to water mass changes in the proportion of AABW versus NADW, as AABW water is about 3°C colder than NADW. At the end of the

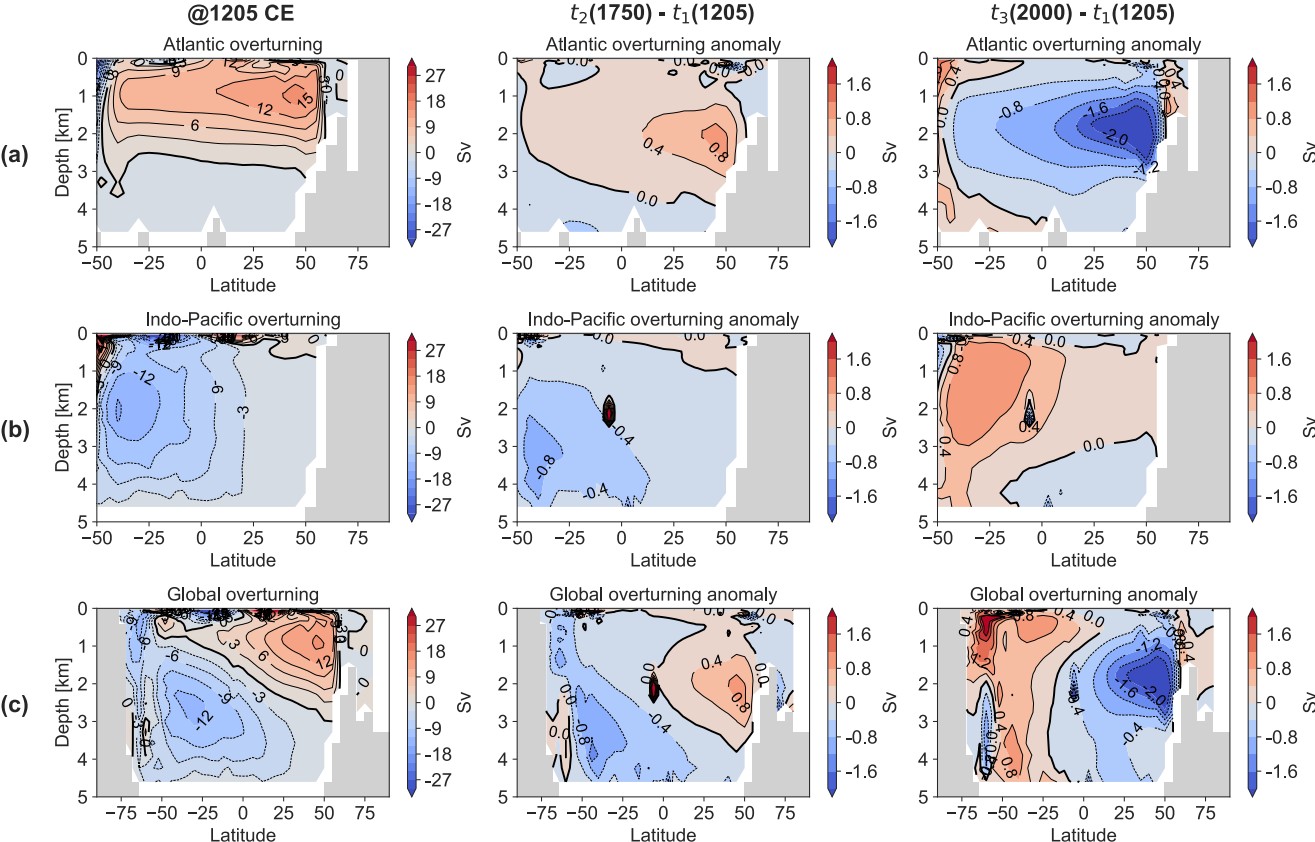

**Figure 3.** Meridional overturning circulation for the transient ocean-only simulation OcTRA for **(a)** the Atlantic (AMOC), **(b)** the Indo-Pacific (IPMOC) and **(c)** globally at different time steps: at the start (1205 CE) and the anomalies at the coldest point of the Little Ice Age (1750 CE) and in 2000 CE. Values are averages over 15 years (for 1205-1220 CE) respectively 30 years (around 1750 and 2000 CE). Note the changing colorbar and latitude axis. The overturning stream function is positive (red) for water rotating clockwise and negative (blue) for counterclockwise.

simulation (2800 CE), relatively more cold AABW at the expense of NADW is observed at depths from 2 to 4 km, which corresponds to the location of the persistent cold anomaly (Fig. A3). The opposite (an increase in NADW at the expense of AABW) occurs above 2 km and below 4 km, explaining the stronger Atlantic warming in TRA at these depths. In the Pacific, the more effective downward transport in TRA is explained by increased inflow of deep southern water during the cooling and afterwards the positive IPMOC anomaly causes less inflow of cold AABW during the subsequent warming and hence a more

effective warming at depth in TRA than in FIX (see Fig. 3). The SO is already warming in year 2000 for both TRA and FIX, because of its high ventilation rate.

The red lines in Fig. 4a and b track the minimum temperature anomaly at every depth. This helps us identify whether the deep ocean at present (2000 CE) is still cooling (red line to the right of vertical line), or already warming. Leads and lags

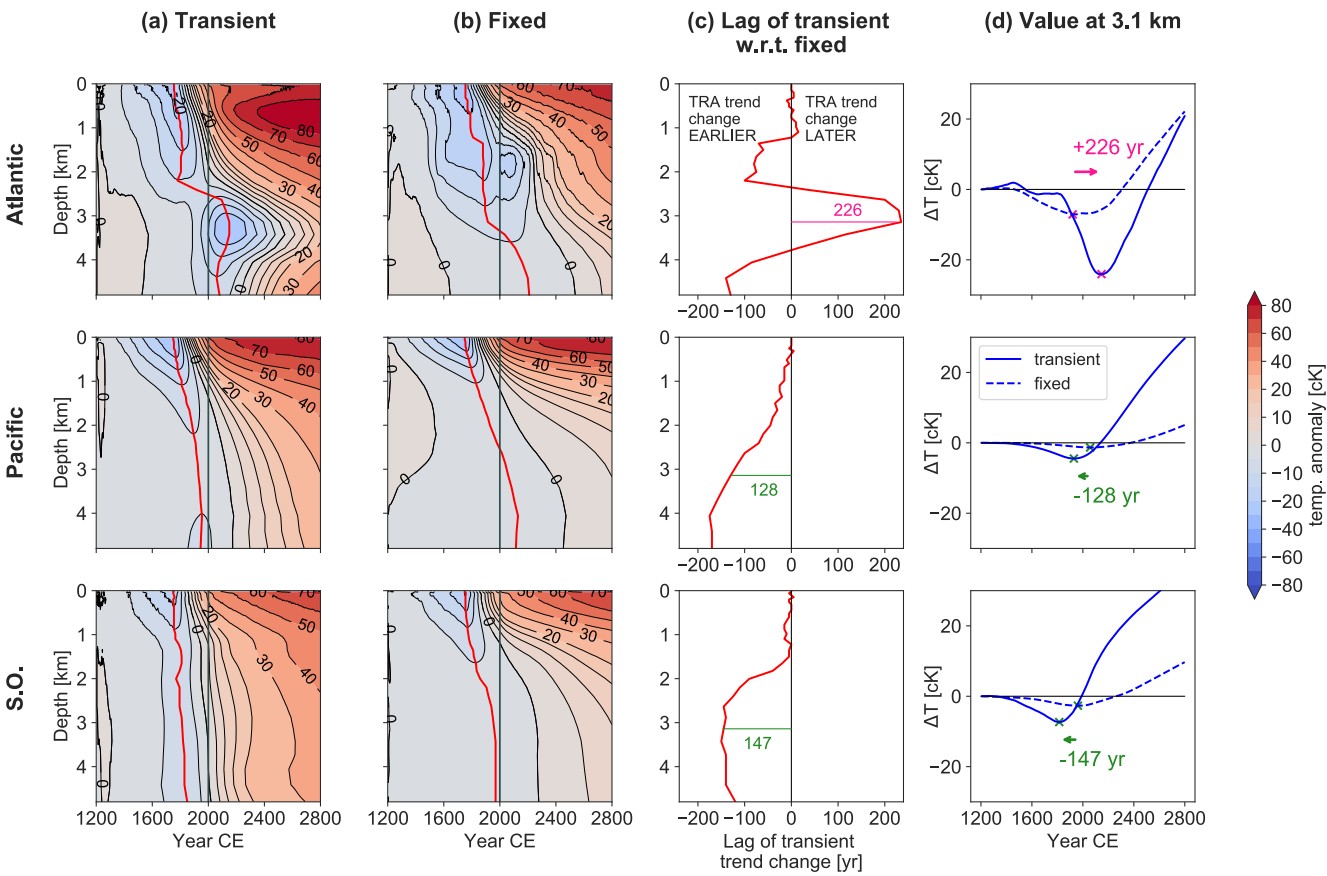

**Figure 4. (a), (b)** Hovmöller diagrams of temperature anomalies in centi-Kelvin (1 cK = $10^{-2}$ K) over depth. Values are basin averages of the Atlantic or Pacific ($35°$S–$70°$N) or the Southern Ocean ($< 35°$S), respectively. Colour levels increase in steps of 5 cK from –20 cK to 20 cK and in steps of 10 cK otherwise. Red lines track the minimum temperature anomaly and indicate where the temperature trend changes from cooling to warming. **(c)** The lag in occurrence of the transient minimum temperature anomaly w.r.t. the fixed. This equals the red line in (a) minus the red line in (b). **(d)** Temperature anomalies from (a), (b) are repeated for a fixed 3.1 km depth. Pink and green arrows indicate lags (positive) and leads (negative) of the transient w.r.t fixed simulation in the temperature trend change at this depth (also indicated in (c)). Results are for simulations OcTRA (transient) and OcFIX (fixed).





of the temperature minimum of the two simulations FIX and TRA can be readily determined. Figure 4c provides the depth
dependence of the lag of the temperature minimum in TRA with respect to that in FIX. The TRA minimum in the Pacific and
the SO appears earlier and this lead grows up to 150 years at depth. In the Atlantic, this structure is interrupted by a strong
lag between 2.5 and 4 km of over 200 years. Hence, changing circulation accelerates the propagation of the downward signal
everywhere except in the Atlantic between 2.5 and 3.5 km depth, providing a mechanism for ocean memory of past changes
in surface climate. In Sect. 4, we will carry out sensitivity tests to assess the robustness of our findings.

Figure 4d presents the time series view of temperature anomalies at 3.1 km, the depth where the strongest differences
between the basins are found. Here, changes in ocean circulation delay the arrival of industrial warming by more than 200
years in the deep Atlantic, whereas they accelerate the arrival by about 120 and 150 years in the deep Pacific and deep SO,
respectively. Another notable feature is that the amplitude is higher for TRA than for FIX in each basin. As discussed in Sect.
4, this is a robust feature. Similarly, Xie and Vallis (2011) found that changing ocean circulation increases the effective depth to
which anomalous heat reaches under their warming experiments. They attribute this to changing circulation allowing for more
heat uptake via a) reducing average SST and b) reducing surface heat loss at high latitudes. As both explanations involve SST
differences between OcTRA and OcFIX, which we do not have in our set-up, there must be even more reasons why circulation
changes promote the downward propagation of temperature anomalies.

### 3.3 Decomposition of temperature anomalies

Anomalies of the transport of a quantity $Q = Q(x, y, z)$, $v Q$, in the ocean can be decomposed into contributions originating
from changes in transport velocity $v$ and in the quantity itself. Following Winton et al. (2013), we write:

$$v Q = v_0 Q_0 + v' Q_0 + v_0 Q' + v' Q' \ , \tag{1}$$

where $Q_0$ and $v_0$ are the steady state values of the quantity $Q$ and the transport velocity $v$ (including advection, diffusion and
convection), and $Q'$ and $v'$ are their deviations. Here, $Q$ can be any quantity transported in the ocean, e.g., temperature, salinity,
heat, etc. We consider the heat content density:

$$Q = C_p \cdot \rho_0 \cdot T, \tag{2}$$

where $C_p = 3981 \ \mathrm{J \, kg^{-1} \, K^{-1}}$ is the specific heat capacity, $\rho_0 = 1027 \ \mathrm{kg \, m^{-3}}$ is the reference water density obtained from the
average seawater density in steady state, and $T$ is the potential temperature. With $Q$ as heat content density, the first term on
the right hand side of Eq. (1) represents the steady state heat transport, the second and third terms, $v' Q_0$ and $v_0 Q'$, quantify
anomalous heat transport due to changes in circulation and in SST, respectively. The last term $v' Q'$ is quadratic in the deviations
and hence small. Based on an analysis of global means in a climate change experiment with a 1 % per year increase in $CO_2$
and evaluated at year 100, Winton et al. (2013) give an estimate of $v' Q_0 : v_0 Q' : v' Q' = 1 : 0.3 : 0.1$.

We can find the contribution of each term in Eq. (1) by making use of experiments OcTRA ($v' \neq 0$) and OcFIX ($v' = 0$):
$v_0 Q_0$ is calculated from OcTRA or OcFIX at year 1200; $v' Q$ from the difference between the two, i.e., OcTRA – OcFIX;
and $v_0 Q'$ results from the difference of OcFIX and its steady state, i.e., OcFIX – OcFIX($t = 1200$). The sum of all these

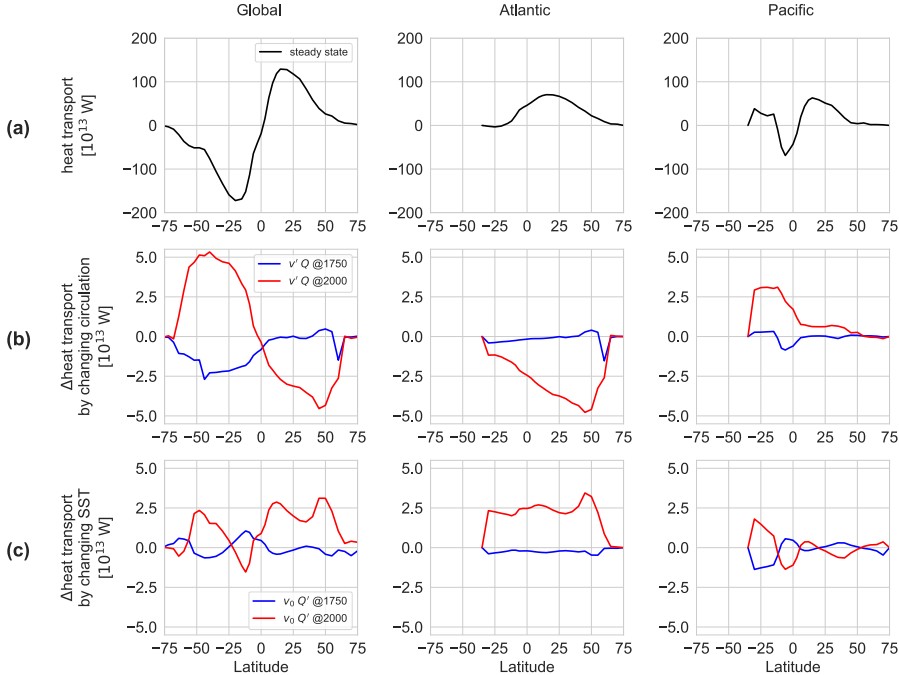

**Figure 5. (a)** Global and basin northward meridional heat transport in the ocean at steady state, OcFIX($t = 1200$). Meridional heat transport anomaly due to **(b)** changing circulation, and **(c)** changing SST. The coldest time of the LIA is shown in blue (1750 CE) and the industrial warming phase in red (2000 CE). Note the different order of magnitude on the y-axis. Results are derived from simulations OcTRA and OcFIX (see text) and variables $v$ and $Q$ refer to Eq. (1).

indeed corresponds to OcTRA, which equals $v\,Q$. As mentioned by Winton et al. (2013), it is not possible to separate the terms $v'Q = v'Q_0 + v'Q'$, because the model only exhibits changing circulation $v'$ under $Q' \neq 0$ as otherwise surface density is unchanged. A difference between this study and Winton et al. (2013) is that in our setup $Q'$ is the same for fixed and for transient simulations, since it is determined by the prescribed SSTs, which are identical for OcFIX and OcTRA.

The decomposition of the northward global heat transport $v\,Q$ is shown in Fig. 5. The response of the meridional heat flux during the medieval cooling occurs primarily in the southern hemisphere and is caused by a strengthening of the circulation in the SO and Indo-Pacific and hence a stronger southward transport with only small changes in the Atlantic basin. The industrial warming is characterized by a decreasing poleward heat flux in both hemispheres. In the south, this is primarily due to changes in the Pacific circulation and changing SST and circulation in the SO. In the northern hemisphere, the weakening poleward heat

transport is due to changes in the Atlantic Ocean. Here, changing circulation dominates in the form of a weakening overturning, which is only partly compensated by anthropogenic warming of the upper ocean.

We compare the meridional heat flux anomaly due to changing circulation during the industrial warming phase (Fig. 5) to Winton et al. (2013), their Fig. 5 (bottom, black line). The shape of the redistribution heat transport in our Fig. 5b is comparable





to Winton et al. (2013), but it is about an order of magnitude smaller. In the warming experiment of Winton et al. (2013), the
radiative forcing is $4.5\,\mathrm{Wm^{-2}}$, whereas the radiative forcing in our experiment is only $1.4\,\mathrm{Wm^{-2}}$.

In addition to heat transport, the column-integrated heat content, hereafter called ocean heat density (OHD), informs about
the distribution of heat uptake. OHD is given as

$$\mathrm{OHD} = \int_{-H}^{0} C_p\,\rho_0\,T\,dz, \tag{3}$$

where $H$ is the total height of the water column. OHD anomalies in year 1750 and 2000 are shown in Fig. 6. The medieval
cooling causes a decrease in OHD globally (except in the Arctic in TRA), with relatively homogeneous cold anomalies in
TRA, whereas FIX possesses the strongest cooling in the Atlantic in year 1750. For FIX, the 550 years of cooling were not
enough to reach the old waters in the deep Pacific, but in TRA stronger Pacific anomalies are caused by circulation changes,
which overlay the steady state transport of anomalous heat. In 2000 CE compared to 1205 CE (Fig. 6b,e), FIX still bears the
signature of the LIA in the Atlantic, Arctic and SO. In TRA, this heat deficit has disappeared with the exception of a small
region in the North Atlantic. It is evident that the transient response of the circulation determines the magnitude and regional
distribution of the memory of past sea surface temperature anomalies.

Comparing OHD in 2000 CE to 1750 CE instead (Fig. 6c,f) and hence only considering anomalies during the warming
period, gives a different picture. All heat anomalies are much larger, since the LIA cooling is not subtracted in this case. In
TRA the OHD anomalies are larger than in FIX, confirming once more our finding that signals are propagating more effectively,
i.e., with a larger amplitude, into the ocean in TRA than in FIX. Further, hardly any negative anomalies are present anymore
in TRA and FIX in 2000 when taken with respect to 1750. That is, the warming hole in TRA (Fig. 6b) mainly originates as a
residual from the 1200-1750 LIA cooling and is not due to the AMOC slowdown from 1750 to 2000 in our ocean-only setting.

The global ocean heat content (OHC) is given by $\mathrm{OHC} = \int \mathrm{OHD}\,dx\,dy$. The contributions of circulation and SST changes
are determined by taking the difference between OcTRA and OcFIX respectively taking the anomalies of OcFIX with respect
to its steady state. The result is given in Fig. 7. The decrease of OHC during the LIA is clearly dominated by changing SST,
which globally has an effect about three times larger than changing circulation. During the warming, changing circulation and
changing SST have a similar influence on a global scale with a slight dominance of the former before about 2300 CE; afterwards
changing SST takes over to some extent, because the effect of changing circulation decreases with recovering AMOC. The time
evolution of the partitioning of OHC into circulation and SST related anomalies is different in the two basins (Fig. 7b,c). The
medieval cooling of OHC is completely dominated by SST changes in the Atlantic, whereas the Pacific features an about equal
contribution of circulation and SST related changes. The two effects contribute significantly in both basins under industrial
warming, with a larger component of changing SST in case of the Atlantic. As expected, changing SST contributes in both
basins to an OHC decrease under LIA cooling and an increase under global warming. In contrast, changing circulation has
an unexpected effect in the case of the Atlantic under the LIA cooling: it slightly counteracts the global cooling. This is also
visible in Fig. 6, where the Atlantic cooled more in OcFIX (panel d) than in OcTRA (panel a).





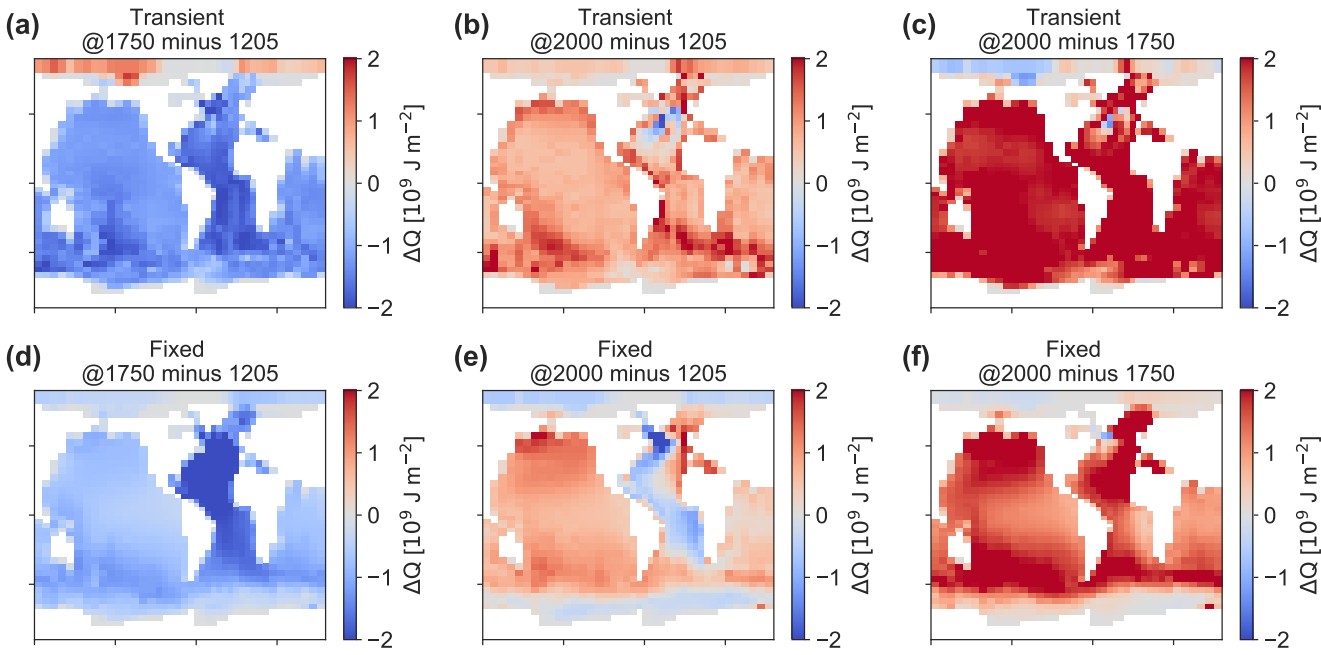

**Figure 6.** Ocean heat density (OHD) anomalies for transient and fixed circulation. **(a), (b)** Anomalies in 1750 and 2000 CE, both with respect to year 1205 for case OcTRA and **(d), (e)** for case OcFIX. **(c), (f)** Anomalies in 2000 CE are also shown with respect to year 1750.

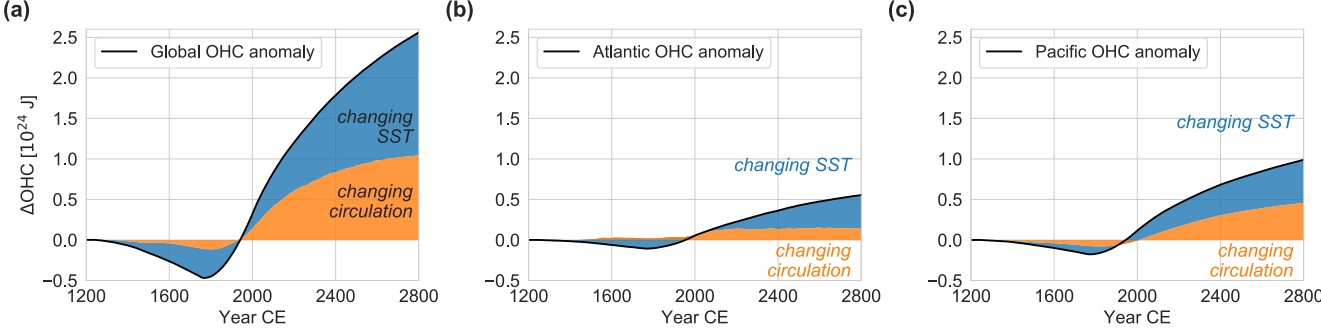

**Figure 7.** Ocean heat content anomaly decomposed in contributions due to changing SST and due to changing circulation for **(a)** the global ocean, **(b)** the Atlantic and **(c)** the Pacific basin, derived from simulations OcTRA and OcFIX.





**Table 2.** Quantification of how different effects act on the deep ocean temperature at 3.1 km, per basin and at times of interest 1750 CE and 2000 CE. Values are temperature anomalies w.r.t. 1200 CE in centi-Kelvin (cK) and percentages are added in parentheses, where the residual is excluded, i.e., 100 % corresponds to the sum of effects 1, 2a, 2b and their interaction. For the percentage, the absolute value of each contribution is taken. Calculations and uncertainties are explained in Appendix B.

| Multiple effects @3.1 km on: | | 1. Changing SST 1. History of SST | 2. Changing circulation[a] 2a. Water masses | 2b. Water age | Interaction[b] | Residual[c] | Total[d] |
|---|---|---|---|---|---|---|---|
| Atlantic: | 1750 | −4.9 (57%) | 3.5 (41%) | −0.17 (2%) | −0.004 (0%) | 0.18 | −1.3 |
| | 2000 | −3.3 (32%) | −2.0 (19%) | −5.0 (48%) | −0.025 (0%) | −3.7 | −14.1 |
| Pacific: | 1750 | 0.011 (1%) | −1.1 (97%) | 0.023 (2%) | −0.0001 (0%) | −1.5 | −2.6 |
| | 2000 | −0.46 (26%) | −1.3 (72%) | −0.038 (2%) | −0.002 (0%) | −2.2 | −4.0 |
| Southern Ocean: | 1750 | −0.97 (25%) | −2.9 (74%) | −0.059 (1%) | −0.006 (0%) | −2.5 | −6.5 |
| | 2000 | −6.1 (73%) | 2.2 (26%) | 0.008 (0%) | −0.006 (0%) | 4.6 | 0.72 |

[a] Changing circulation consists of the – not necessarily independent – subeffects 2a and 2b.

[b] The interaction is the cross term of deviations between effect 1 and 2a.

[c] The residual equals the total minus effects 1, 2a, 2b and the interaction. It represents unaccounted effects and uncertainties in the calculations.

[d] The total of all effects is given by the temperature anomaly in the OcTRA model output, which is averaged at 3.1 km within the basin of interest.

In the previous, we decomposed heat transport and OHC into the contributions of changing circulation and changing SST. Recall that the Hovmöller diagrams in Fig. 4a, b presented basin-averaged temperature anomalies over depth. These temperature anomalies, which are directly related to OHC (Eq. (3)), can readily be decomposed as well: the OcFIX anomalies in Fig. 4b already equal the contribution of changing SST, and the contribution of changing circulation is found by subtracting column OcFIX from OcTRA. The result (Fig. A2) confirms that changing SST causes the expected downward propagation of cooling respectively warming, whereas changing circulation causes the persistent cold anomaly in the Atlantic around 3 km depth with an enhanced warming above and below this depth in addition to an enhanced warming after 2000 CE in the Pacific and SO.

### 3.4 Explaining the leads and lags

Now we turn our attention to deep ocean temperature and the effects acting on it in a more local framework: our aim is to gain more understanding of the leads and lags that we observed at 3.1 km depth in the different basins in Fig. 4. Instead of a true decomposition, we present a rough estimate of each contribution. Moreover, we split the effect of changing circulation in two subeffects: shifts in water mass distribution, which determine whereto regional SST anomalies are carried, and the absolute overturning rate, i.e., how fast deep ocean water is replenished. The results are summarized in Table 2 and explained below.

Firstly, historical SST anomalies differ between regions such that deep water formation regions experience different surface temperature trends. As an illustration, we refer back to Fig. 1b for our idealized forcing. Recall that regions are defined in Fig. A1 and correspond to the source regions of different dye tracers. We notice relatively high amplitudes in the North Pacific and South Atlantic, and low amplitudes in the Arctic, but these regions play a minor role in determining the ocean temperature at depth. The larger variability of northern North Atlantic SST is due to convection occurring in only 40 grid cells there. This



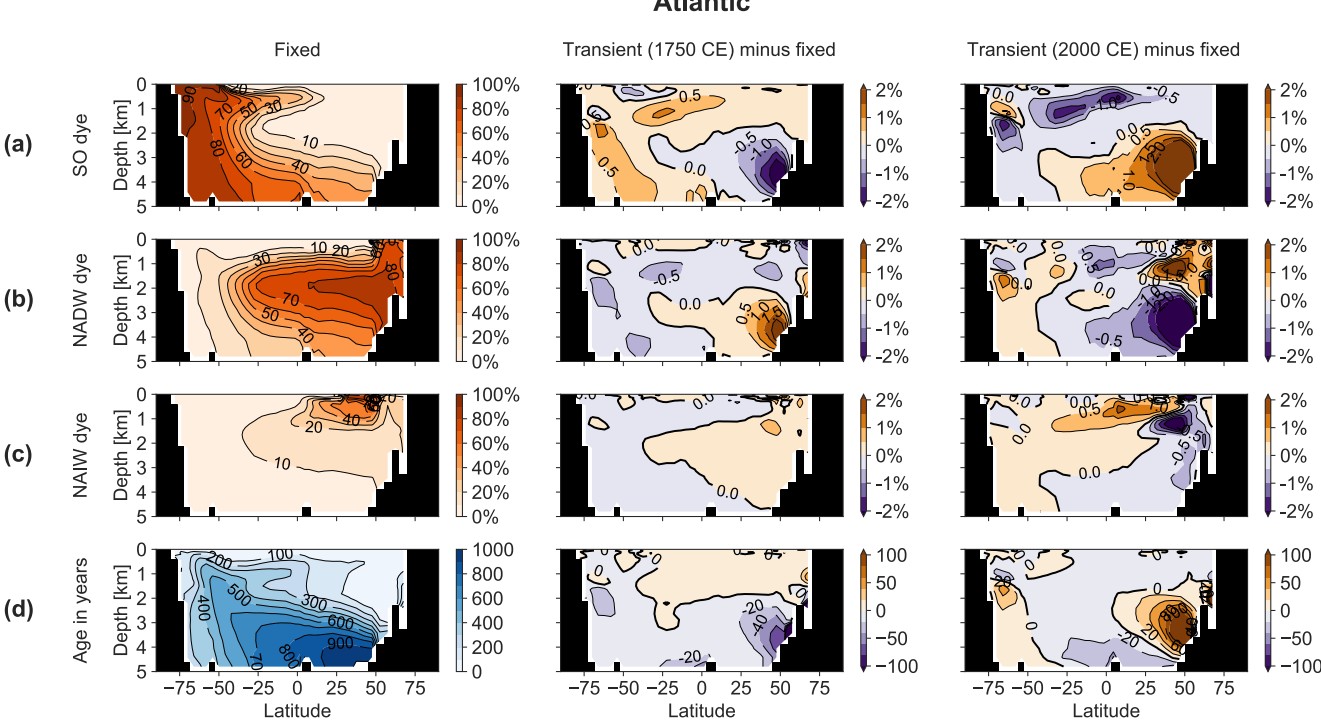

**Figure 8.** Zonally averaged meridional section of the Atlantic basin, including the SO sector, showing **(a)** Southern Ocean dye tracer, **(b)** North Atlantic Deep Water dye tracer, **(c)** North Atlantic Intermediate Water dye tracer and **(d)** ideal age. Columns show the fixed-circulation simulation (OcFIX) and the anomaly of transient (OcTRA) w.r.t. fixed in 1750 (coldest time in LIA) and 2000 CE. Units for dye are % of initial dye surface concentration.

region's SST (NADW formation) sees no clear cooling during the first two centuries, but catches up with the global average
cooling trend afterwards. The Southern Ocean SST (AABW and AAIW) starts cooling rapidly, but with a smaller amplitude
than the global average. Comparing these two most important water masses, we conclude that NADW sees less of the LIA
cooling than AABW during 1200-1400 CE, but more during 1500-1750 CE. The effect of SST history (effect 1) at a fixed 3.1
km depth is computed (Table 2), where the SST history of all 8 water masses is taken into account (Appendix B). Evidently,
effect 1 contributes a cooling in 1750 CE in the deep Atlantic and SO, but no SST-caused cooling reaches the deep Pacific yet.
In 2000 CE, the cooling contribution reached the deep Pacific and is still present at this depth in all basins.

Let us turn to effect 2a, shifts in the distribution of water masses, which are illustrated by means of dye tracers. The merid-
ional distribution of the water masses that dominate the abyss is shown in Fig. 8a-c (Atlantic) and Fig. 9a-c (Pacific). We track
the combined water masses of AABW and Antarctic Intermediate Water (AAIW) with one Southern Ocean dye (SO). First we
discuss the changes in 1750: the largest water mass changes occur in the North Atlantic at 3 km depth, where the NADW frac-
tion increases by about 2 percent points at the expense of AABW. This has a warming effect that counteracts the LIA cooling in



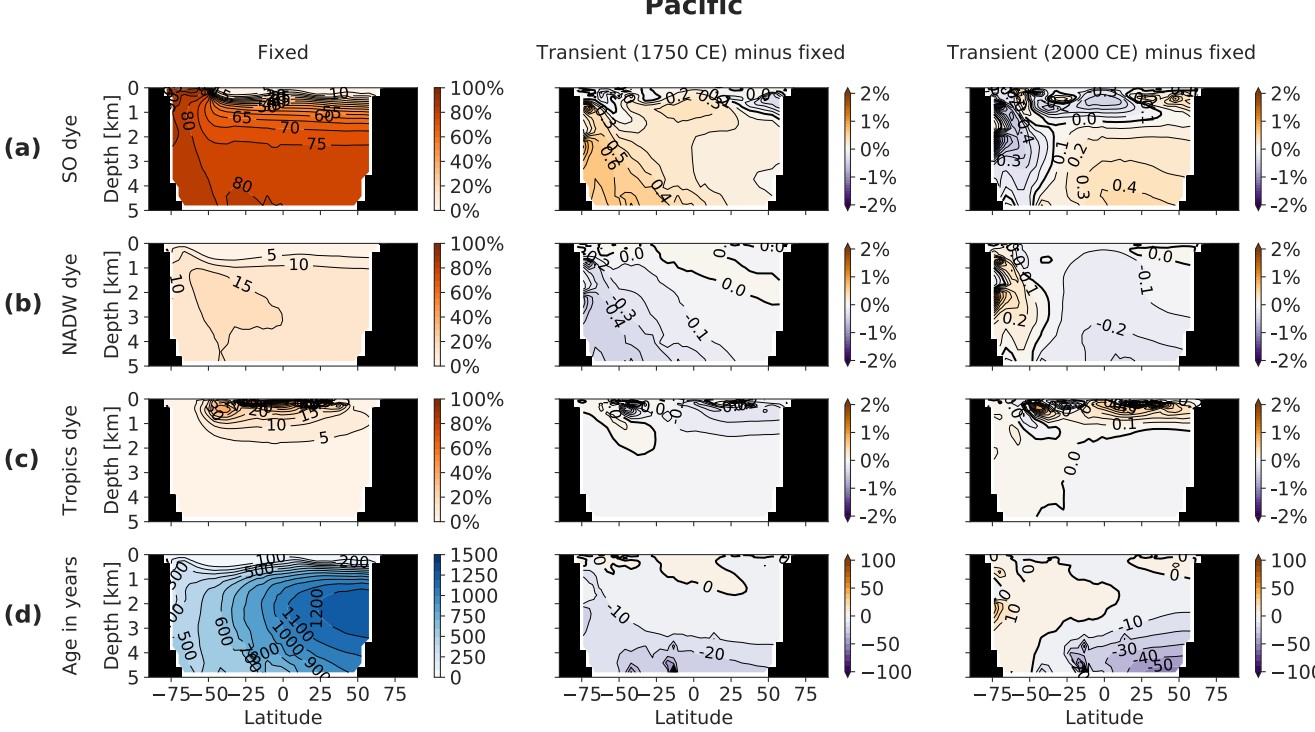

**Figure 9.** Zonally averaged meridional section of the Pacific basin, including the SO sector, showing **(a)** Southern Ocean dye tracer, **(b)** North Atlantic Deep Water dye tracer, **(c)** tropics dye tracer and **(d)** ideal age. Columns show the fixed-circulation simulation (OcFIX) and the anomaly of transient (OcTRA) w.r.t. fixed in 1750 (coldest time in LIA) and 2000 CE. Units for dye are % of initial dye surface concentration. Colour levels for Pacific anomalies increase in steps of 0.1 % dye or 10 years age, respectively.

the Atlantic. In the deep Pacific however, slightly more AABW flows in from the south at the expense of NADW. Analogously, this has a cooling effect and enlarges the propagation of the LIA cooling in the transient case. Table 2 confirms that effect 2a, shifting water masses, leads to a warming in the Atlantic and a cooling in the Pacific at 3.1 km. Note that here all water masses are taken into account such that a second-order effect is present: relatively more deep water formation water masses (NADW and AABW) relative to other water masses under the LIA forcing, which leads to an additional cooling. In the year 2000, we see a cooling in the Atlantic and Pacific at this depth, whereas the SO warms. Dye tracers in the Atlantic reveal a pattern that is opposite to 1750, so they track a cooling at 3.1 km depth. Water mass anomalies in 2000 south of 35°S show that the SO warming is caused by more NADW-sourced water and less local SO water. It is surprising that in the Pacific (excluding the SO sector), the same qualitative water mass anomalies as in 1750 are still present, however displaced northwards, and opposite trends are developing in the SO. This is because of a delay: the amount of SO water mass is already decreasing in 2000 (and NADW increasing) but the decrease initiated only around 1900 CE, such that the water mass anomaly built up over 1200-1900 CE is not compensated for yet (not shown).



Effect 2b is the absolute overturning rate, estimated by changing water age. In principle it is possible that the rate at which water masses refresh would change without any change in the relative composition of water masses. For instance, AABW

and NADW would then still be present in the same proportion, but a general speed-up of the circulation (a faster overturning) would make the water age younger. Intuitively, a strengthening of the global MOC brings more cold polar surface water to the abyss resulting in a global cooling of the deep ocean, because downward heat diffusion from low-latitude surfaces is opposed by upwelling of colder deep water. From the ideal water age in 1750 CE in Fig. 8d and Fig. 9d, we see that the water becomes younger by up to ∼40 years in the deep Atlantic and up to ∼10 years in the deep Pacific at 3 km depth. In 2000 CE, the age

anomalies are in the other direction, as expected: the deep North Atlantic ages by more than 100 years, whereas the Pacific SO becomes ∼5 years older. The rest of the Pacific continues to become younger, which is again due to a delay and the water will become older, with respect to 1200 CE, in 2100 CE (not shown). We conclude that effect 2b causes the deep Atlantic to see an older, hence warmer, signal during the LIA cooling and a younger, hence colder (noting that the ∼500 yr old water in 2000 CE still sees SSTs of 1500 CE), signal during the warming period after 1750. This contributes to a delay during both periods.

We expect that the influence in the Pacific and SO is minor because of the small changes in age. For effect 2b, many values in Table 2 have different signs than expected based on the above analysis of ideal age changes. We conclude that the values indicated as ≤ 2 % are indeed not significantly different from zero. Thus, effect 2b has a very small impact at 3.1 km, except in the case of the Atlantic in 2000, where we also observed the largest change in ideal age.

Finally, we assess the relative contributions of effects 1, 2a and 2b at 3.1 km (Table 2). Surprisingly, the change in absolute

overturning rate, which is estimated via water age, has practically no influence in all cases except the Atlantic in 2000 CE. In all other cases, deep ocean temperature anomalies are determined by the interplay of SST and water mass changes. In the Atlantic, SST and water mass changes contribute roughly equally in 1750 but with opposite signs. Therefore, the shifted water masses cause a delay in the arrival of LIA cooling at depth until 1750. In 2000, both water masses and water ages now strongly enhance the amplitude of the cooling, which arrives relatively late. As such, the arrival of the industrial warming is delayed as

well. These mechanisms explain the two-century lag observed in the Atlantic and nicely correspond to Fig. 4d, where we see the amplitude of TRA dampened until 1750 but enhanced until 2000 (and further). In the Pacific and SO, the contribution of water mass changes generally dominates over SST (except for the SO in 2000). These water mass changes contribute with the same sign as the forcing such that they fasten and enhance the arrival of anomalies at depth. This explains the leads observed in the Pacific and SO and the larger amplitudes in TRA.

## 4  Sensitivity of leads and lags

The leads and lags of temperature minima from the LIA cooling (Fig. 4c, d) are determined by model-dependent changes in circulation, including advection, diffusion and convection. In this Section, we test how robust the leads and lags are under varying key model parameters for mixing and wind. We run additional transient and fixed simulations with halved and doubled mixing, respectively wind stress (Table 1). The changes are applied globally and uniformly. Figure 10 summarizes the resulting

leads (negative) and lags (positive) throughout the water column of TRA with respect to FIX simulations.



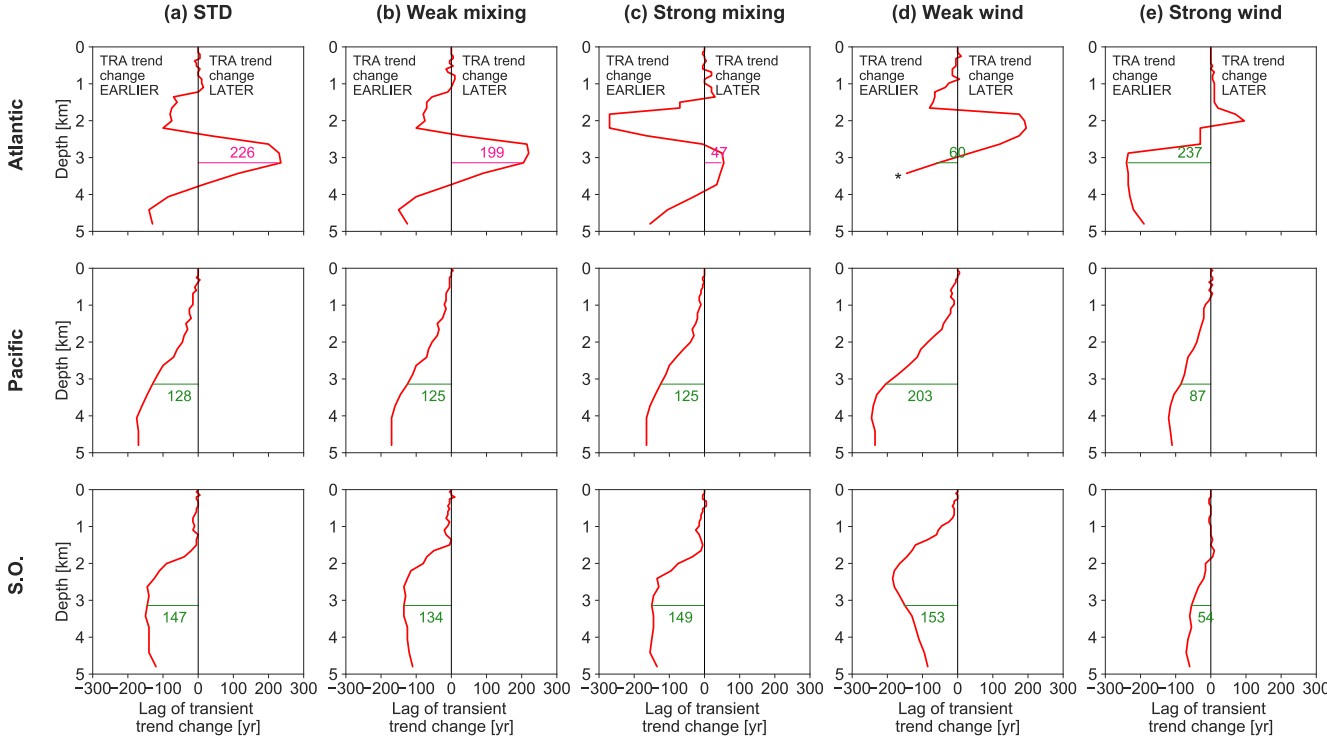

**Figure 10.** Lags (positive) or leads (negative) in the arrival of the warming at each depth (transient minus fixed simulation). Results are shown for **(a)** the standard simulations OcTRA and OcFIX (repeated from Fig. 4c), **(b, c)** simulations with varying mixing (OcTRA_weakmix and OcFIX_weakmix respectively OcTRA_strongmix and OcFIX_strongmix) and **(d, e)** varying wind stress (OcTRA_weakwind and Oc-FIX_weakwind respectively OcTRA_strongwind and OcFIX_strongwind). Numbers indicate the values of leads and lags at 3.1 km depth. *For (d), the red line stops in the Atlantic below 3.4 km, because the LIA cooling anomaly is not detectable in OcFIX_weakwind at these depths such that its trend change is undefined.

An overall picture appears of leads in the Atlantic that are interrupted by a depth range of lags, and leads throughout the Pacific and SO. These lags in the Atlantic are located between about 2.5 and 4 km for standard, weak mixing and strong mixing simulations, but they are shifted upwards for weak and strong wind, from 1.5 to 3 km and from 1.5 to 2 km, respectively. Therefore, our finding that changing circulation delays the propagation of the downward signal in a certain depth range in the

Atlantic but accelerates it everywhere else is robust, with the precise depth range boundaries depending on model parameters.

At a fixed 3.1 km depth, the leads and lags are relatively robust under varying mixing, but not under varying wind, especially in the Atlantic. The diapycnal diffusivity $K_d$ is halved or doubled to vary mixing (Fig. 10b, c). In the Atlantic, weak mixing has not much influence, but strong mixing makes the ~226 yr lag almost disappear. This is explained by the presence of two local temperature minima in the Atlantic for OcFIX (Fig. 4b), OcFIX_weakmix and OcFIX_strongmix. Only in the case of

OcFIX_strongmix, the global minimum, which is marked to compute the lag, is shifted towards the second local minimum. A change in FIX compared to the standard simulation OcFIX is possible at all, since the steady states for weak and strong mixing





are different (Fig. A4-A5). Lead times in the Pacific and SO stay within the range of 125 to 150 years and are hence insensitive to these rather large changes in $K_d$. Steady states possess especially different MOCs for halved and doubled wind stress $\tau_{wind}$. Increasing wind stress almost doubles the MOC in the northern hemisphere and creates a strong Deacon cell in the SO, i.e.,
Ekman-driven equatorward flow. This causes large differences in simulated leads and lags for varying wind (Fig. 10d, e). The lag in the Atlantic becomes a lead instead, whereas the leads in Pacific and SO range from about 50 to 200 years.

For the standard simulation, amplitudes of LIA cooling were considerably larger in TRA than in FIX at 3.1 km depth, as noted previously in Fig. 4d. This property is robust under varying mixing and wind: all sensitivity simulations possess larger TRA than FIX amplitudes at 3.1 km in all basins, except strong wind in the Atlantic (not shown). Thus, circulation changes
enhance the amplitude, hence detectability, of deep ocean temperature anomalies.

## 5   Comparison to Gebbie and Huybers (2019)

Our study was motivated by the earlier study of Gebbie and Huybers (2019) (GH19, henceforth), who interpreted deep ocean temperature anomalies with an inverse model with fixed circulation, inferred from oceanographic measurements from the late 19$^{\text{th}}$ century and modern observations. Early transects in the Atlantic and Pacific oceans are available from the expedition of
the HMS *Challenger* of 1872-1876, and modern data were obtained during WOCE in the 1990s. GH19 have identified coolings and warmings at a depth range of 1800 to 2600 meters, which they attribute to remnant deep ocean signals of surface climate change. They find that the deep Pacific has been cooling during that period, but the deep Atlantic, particularly in the northern hemisphere, has been strongly warming.

We refer back to the basin-averaged Hovmöller diagrams in Fig. 4a-b. These compare directly to Fig. 1b-c from GH19,
though with a different time axis: we show $1200 - 2800$ CE, whereas they simulate $0 - 2000$ CE, including a medieval warm period around 600 CE. The downward propagation of temperature anomalies in the Pacific qualitatively agrees between the two studies. The result of GH19 in the Atlantic is qualitatively similar to our FIX case but not to TRA, which is as expected since their model has a fixed circulation. Quantitative amplitudes of deep ocean cooling until 2000 CE are similar in the Pacific in the two studies, but GH19 show slightly stronger cooling in the Atlantic (up to 20 cK instead of 15 cK), which also extends
deeper.

Now we answer the question whether in the simulations the deep ocean (2 km and deeper) at present is still cooling from the LIA or already warming. Recall that the red lines in Fig. 4a-b help us identify whether the deep ocean at present (2000) is still cooling (red line to the right of 2000) or already warming (red line to the left of 2000). GH19 predicted that the deep Pacific is still cooling at present, but the deep Atlantic already warming. For the Pacific, this is consistent with our FIX case
but not with TRA, which is thought to be more realistic. In our study, most of the deep Atlantic water column is still cooling at present for both TRA and FIX (although the $2 - 3$ km part is already warming in FIX). This substantial difference between our study and GH19 in the Atlantic occurs because the deep North Atlantic is too old in the Bern3D ocean model compared to radiocarbon observations (Gebbie and Huybers, 2012), which correspond to the ages in the observation-based model of GH19.

These observations show radiocarbon ages of up to 600 years in the North Atlantic (zonal average), whereas we find ideal ages
over 900 years (Fig. 8). Therefore, our model needs more time to detect the warming in the deep Atlantic.

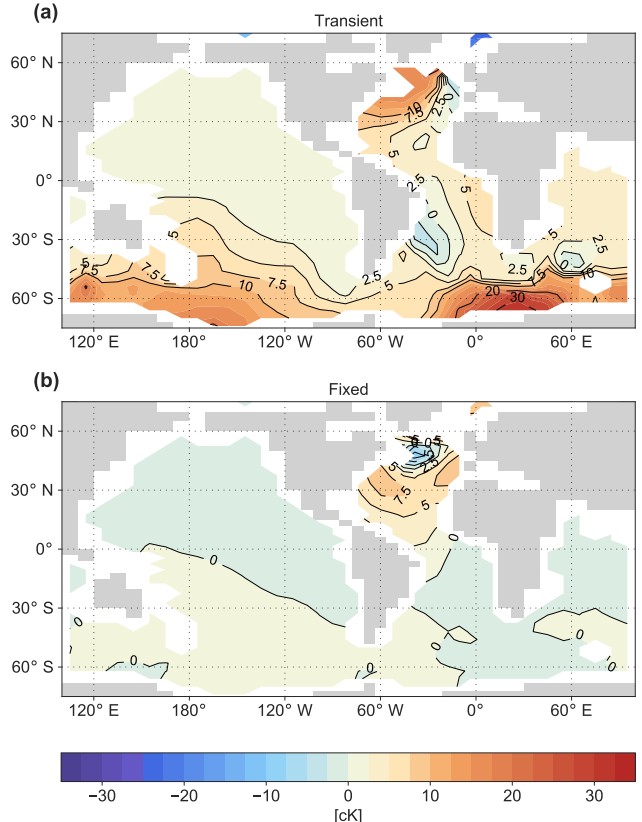

**Figure 11.** Reproduction of Fig. 2 of Gebbie and Huybers (2019) for the two cases of transient and fixed circulation. Deep ocean temperature anomalies are determined between year 1875 (*Challenger* expedition) and 1995 (WOCE campaign) and averaged over the depth of 1912 to 2520 m, for simulations **(a)** OcTRA (transient) and **(b)** OcFIX (fixed). For comparison, see Fig. 2 of Gebbie and Huybers (2019).

In addition, we reproduce Fig. 2 from GH19 for the two cases of OcTRA and OcFIX, which focuses on a depth of about 2 km (Fig. 11). Note that in our TRA case it makes a significant difference which depth we consider, whereas their anomalies travel down more uniformly over depth. At 2 km depth, the fingerprint of circulation changes is not as strong as for instance at 3 km. Figure 11 shows wide-spread warming in both cases in the North Atlantic. However, both cases also exhibit cooling in
some limited areas of the North Atlantic. For TRA, cooling is simulated in the northeast Atlantic, whereas FIX shows cooling in the northwestern part of the North Atlantic. In the Pacific, warming (TRA) or cooling (FIX) trends are very small. In our FIX simulation, cooling is still found in the Pacific, but it is rather weak, and the southwestern Pacific is already warming. The difference between *Challenger* and WOCE data, as well as the model simulation of GH19, show clear remnant cooling in the Pacific in this depth range around 2 km.





The fact that the deep Pacific cooling is weak in our simulations could be caused by the choice of our forcing, which is simpler than that used by GH19 and has a slightly smaller industrial warming amplitude. We have ignored climate variations before year 1200 and start with the medieval cooling from 1200 to 1750. Their simulation starts at 0 CE and has an approximate global mean warming of $0.2°C$ over 600 years with a subsequent cooling. Signals from this early medieval warm period also propagate into the deep Pacific, and we hypothesize that this would be able to delay the arrival of the later cold LIA anomaly

in this region, producing that notable difference.

## 6   Conclusions and outlook

We found that even the relatively small changes in ocean circulation during the last 800 years have a profound impact on deep ocean temperature. While changing ocean circulation fastens the arrival of cold and warm atmospheric temperature anomalies in the deep Pacific and deep Southern Ocean (SO), it delays the arrival of this signal in the deep Atlantic between about $2.5-3.5$

km depth. This lag in the deep Atlantic is two centuries in our simulations and is caused first by a relative increase of NADW at the cost of AABW during the LIA cooling, followed by an older water age under AMOC slowdown during industrial warming. Moreover, circulation changes enhance the amplitude of deep ocean temperature fluctuations in all three major basins, thus making atmospheric anomalies better detectable in the deep ocean.

We have decomposed heat transport and ocean heat content (OHC) anomalies into the contributions of changing circulation

and changing SST. Poleward heat transport decreased under early industrial warming, in which the effect of changing circulation was dominating. Our second decomposition showed that OHC anomalies were for about one third caused by changing circulation during the LIA cooling (hence two thirds by changing SST), whereas both effects had an approximately equal contribution under industrial warming. We conclude that ocean heat uptake is not solely dominated by atmospheric temperature anomalies, but changing ocean circulation has a significant impact.

Lastly, we assessed the robustness of leads and lags under varying mixing and wind strengths. The transient simulation, where ocean circulation adjusts dynamically, still had a larger amplitude than the fixed-circulation simulation in almost all cases and basins. At the fixed depth of interest of 3.1 km, the circulation-caused leads in the deep Pacific and SO versus lags in the deep Atlantic were quantitatively quite robust under varying mixing, but not under varying wind stress. Throughout the water column, the qualitative result of leads in the Pacific and SO is robust under our sensitivity tests. The same holds for the

interesting observation of leads in the Atlantic that are interrupted by lags at a certain depth range (either around 2 or around 3 km).

We are aware that our results are to a certain extent model-dependent, especially on the modeled age of deep waters and on the relative response of AMOC versus SOMOC strength under changing SST. Therefore, we welcome more research concerning the propagation of past surface temperature anomalies to the abyss in other models. A weakness of the coupled Bern3D model

is the representation of atmospheric processes, in particular the hydrological cycle. As such, the generated time-varying SSS field could be replaced in future work by a more realistic SSS from another coupled model or via paleoreconstructions (e.g., the PHYDA database of Steiger et al. (2018)).



In this paper, we held ocean-atmosphere feedbacks and total ocean heat uptake identical between transient and fixed simulations, by prescribing the same SST and SSS time series. We needed this for a correct comparison of the propagation of

(identical) sea surface temperature to depth. In reality, ocean-atmosphere feedbacks and SST patterns are different when the ocean circulation is allowed to dynamically adjust, i.e., in transient simulations (Winton et al., 2013). These effects come on top of the contribution of changing circulation we found. It would be interesting to find a way to combine our approach with the approach of Winton et al. (2013) such that deep ocean temperatures can be compared unbiased without losing the ability of including different ocean-atmosphere feedbacks and studying total OHC change.

In our case study of the past 800 years, we chose an idealized forcing, whereas Gebbie and Huybers (2019) used a forcing with more high-frequency SST fluctuations, which is more realistic but also harder to interpret. Moreover, one could use an even more realistic forcing by prescribing time-varying SST fields with trends that differ between regions, for example with different times of occurrence of the LIA, by obtaining them directly from recent paleoreconstructions (Neukom et al., 2019) instead of from simulation OcAtmTRA. In a more general view, we would be able to understand how any historical SST record

travels down to the deep ocean when we understand the effective depth to which surface temperature anomalies travel (Xie and Vallis, 2011) as a function of the anomaly's frequency and amplitude, and depending on changes in ocean circulation that are a response to this cooling or warming.

*Code and data availability.* The Bern3D model is closed-source, but the output of all model simulations used in this study is available in netCDF format (Scheen and Stocker, 2020). The analysis code that was used to make the figures and to perform the calculations in Appendix

B is available as well-documented python notebook (Scheen, 2020). In the case of questions or suspected bugs, please contact J.S..


## Appendix A: Figures

### A1 General Figures

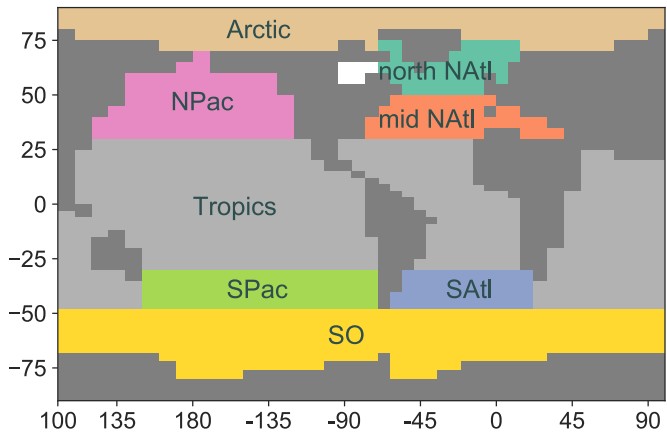

**Figure A1.** Definition of areas as used for regional SST averages on the Bern3D model grid (Fig. 1b, same colors). Simultaneously, these surface areas define where the 8 dye tracers originate (Fig. 8, 9 and A3). The dye tracers are named after the water masses they follow: Arctic, North Pacific Intermediate Water (NPIW, in the area NPac), Tropics, South Pacific Intermediate Water (SPIW, in the area SPac), Southern Ocean (SO), North Atlantic Deep Water (NADW, in the area north NAtl), North Atlantic Intermediate Water (NAIW, in the area mid NAtl) and South Atlantic Intermediate Water (SAIW, in the area SAtl). Together they cover the whole ocean surface except the Hudson Bay. The SO dye tracer contains both AABW and AAIW.



## A2    Figures regarding standard simulations OcTRA and OcFIX

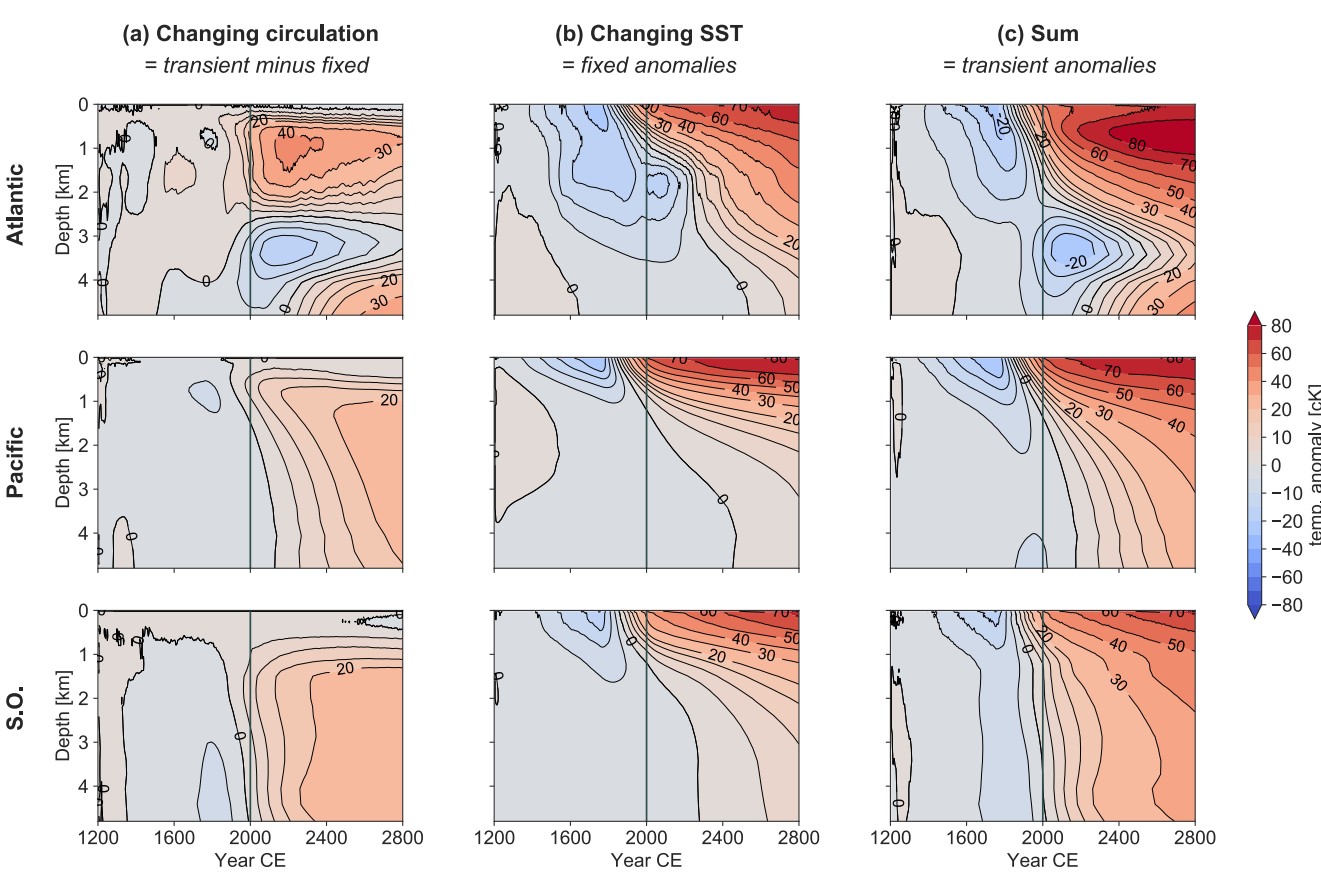

**Figure A2.** Hovmöller diagrams of temperature anomalies over depth, decomposed into the effects of **(a)** changing circulation and **(b)** changing SST with **(c)** the total. Columns (a), (b), (c) equal Fig. 4a minus 4b, Fig. 4b and Fig. 4a, respectively (see main text). Values are in centi-Kelvin (1 cK = $10^{-2}$ K) and are basin averages of the Atlantic or Pacific ($35°$S–$70°$N) or the Southern Ocean ($< 35°$S), respectively. Colour levels increase in steps of 5 cK from -20 cK to 20 cK and in steps of 10 cK otherwise.



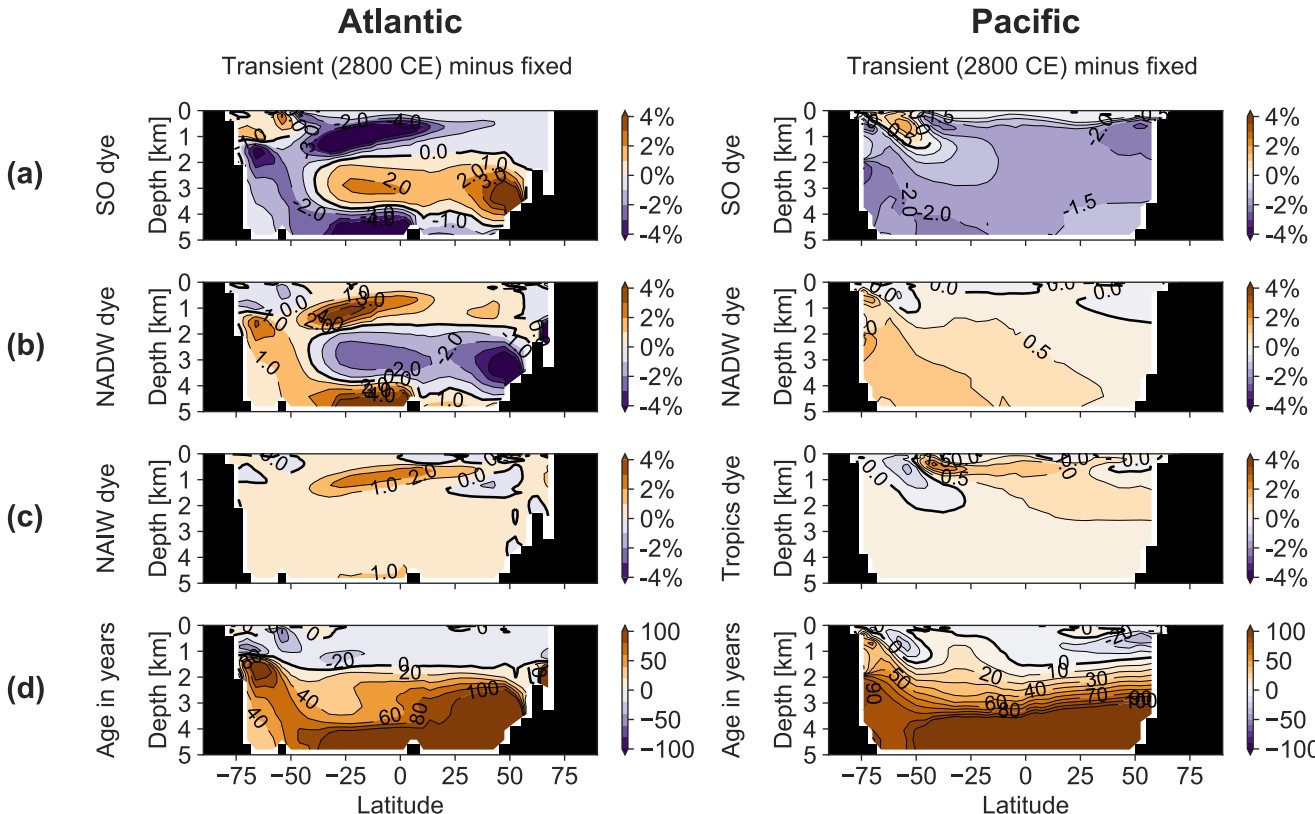

**Figure A3.** Zonally average meridional sections of the Atlantic respectively Pacific basin, including their Southern Ocean sector. Shown are the anomalies of transient (OcTRA) w.r.t. fixed (OcFIX) at 2800 CE (the end of the simulation) of **(a)** Southern Ocean dye tracer, **(b)** North Atlantic Deep Water dye tracer, **(c)** North Atlantic Intermediate Water dye tracer (for Atlantic) respectively Tropics dye tracer (for Pacific) and **(d)** ideal age. Units for dye are % of initial dye surface concentration.

This Fig. A3 shows that the water mass shift between SO dye and relatively warmer NADW in panel (a) and (b) explains
the observed temperature propagation in the Atlantic (Fig. 4a), which shows a persistent cold anomaly between 2 and 4 km combined with enhanced warming above 2 and below 4 km depth. In this argumentation we used the fact that NADW dye tracer also reaches below 4 km, e.g., occupying 10-30 % of the water mass in the South Atlantic at this depth (Fig. 8b). This percentage is larger than expected, because NADW water travelling via the SO back into the deep Atlantic without touching the surface still retains the NADW signature.





## A3 Figures regarding sensitivity-test simulations

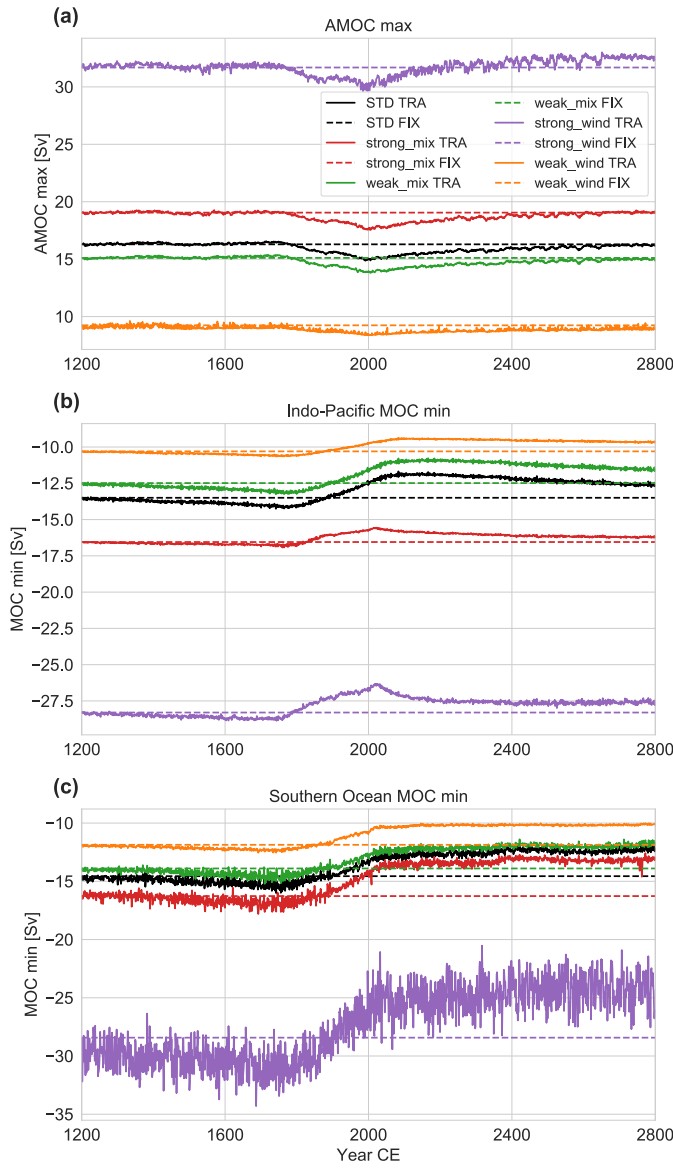

**Figure A4.** For simulations with varying parameters: **(a)** Atlantic Meridional Overturning Circulation (AMOC) maximum, **(b)** minimum of the MOC in the combined Pacific and Indian basin (IPMOC) and **(c)** in the Southern Ocean (SOMOC). Either the wind stress or mixing (i.e., diapycnal diffusivity parameter) is doubled, respectively halved. TRA stands for transient simulations and FIX for fixed-circulation simulations.





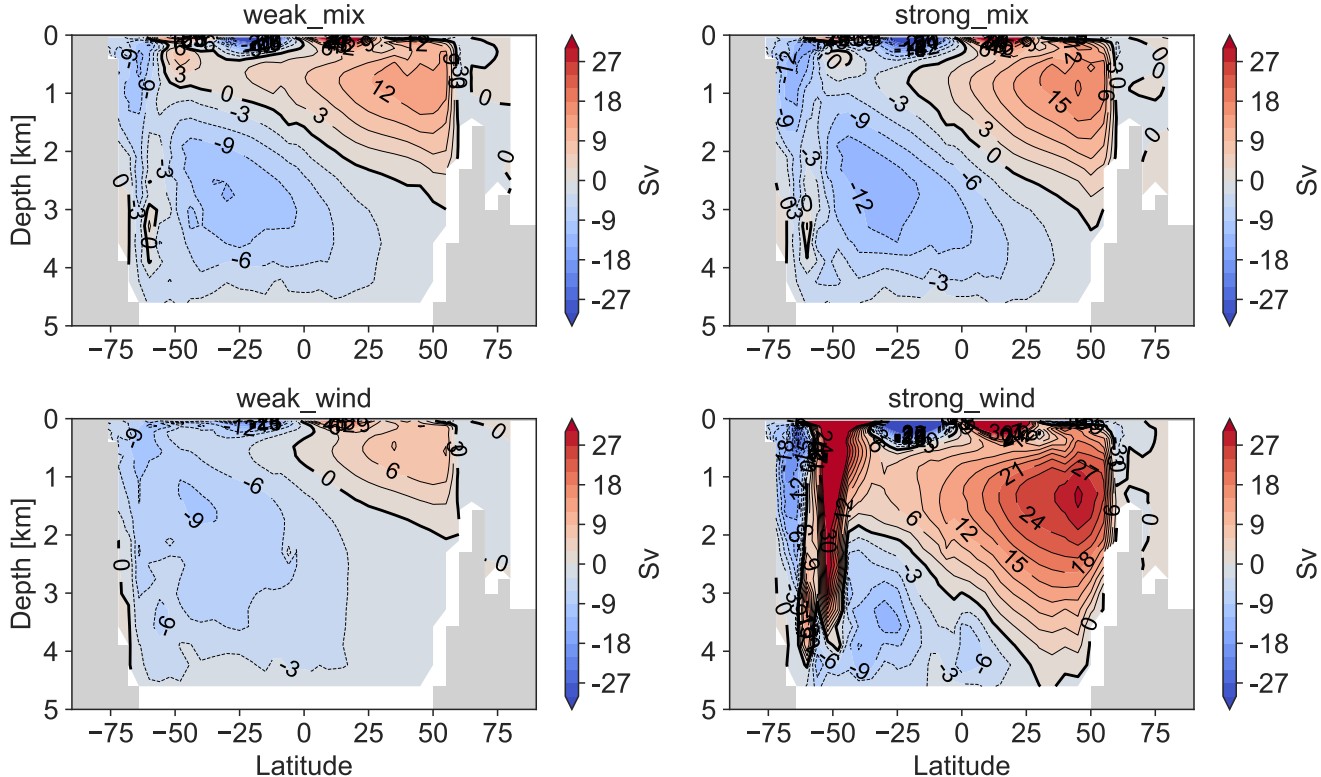

**Figure A5.** Global overturning circulation in steady state (15-year average) for sensitivity simulations (see Fig. 3c for the standard simulations). The overturning stream function is positive (red) for water rotating clockwise and negative (blue) for counterclockwise.





**Appendix B: Quantification of effects causing leads and lags at 3 km**

In this Appendix, we estimate the contribution of multiple effects on the deep ocean temperature at 3 kilometers. This provides more insight in the origin of the leads and lags between TRA and FIX observed at 3 km in the different basins. The end results of these calculations are summarized in Table B3, on which Table 2 of the main text is based.

We are about to quantify the following effects:

1. changing SST

2. changing circulation:

    2a. water masses

    2b. water age

Throughout this Section, variables are only considered at the depth slice of 3.1 km, and within a certain basin, namely the Atlantic, Pacific or Southern Ocean (SO). Typically, variables are computed at steady state, $t_0 = 1200$ CE, or at a certain time step of interest, $t^* \in [1750, 2000]$ CE.

Temperature anomalies $T = T(t)$ at 3.1 km depth can be written in terms of water masses $i$:

$$T(t) = \sum_i T_i(t) \cdot f_i(t),$$

where $T_i(t)$ is the typical temperature of a certain water mass and $f_i(t)$ is this water mass' fraction between 0 and 1. The sum runs over all water masses $i \in$ [NADW, NAIW, SAIW, SO, Arctic, NPIW, SPIW, Tropics]. This corresponds to our 8 dye tracers, which span the entire ocean surface (Fig. A1). Dropping the time dependence and writing $T_i(t)$ and $f_i(t)$ as a sum of their steady state at $t_0$ (subscript 0) and deviation (primed) yields

$$
\begin{aligned}
T &= \sum_i (T_{i,0} + T_i') \cdot (f_{i,0} + f_i') \\
&= \sum_i \underbrace{T_{i,0} \cdot f_{i,0}}_{\text{steady state}} + \underbrace{T_i' \cdot f_{i,0}}_{\text{1. changing SST}} + \underbrace{T_{i,0} \cdot f_i'}_{\text{2a. changing water masses}} + \underbrace{T_i' \cdot f_i'}_{\text{interaction}} \\
&=: T_0 + \Delta T_{\text{SST}} + \Delta T_{\text{WM}} + \Delta T_{\text{int}}
\end{aligned}
$$

This decomposition is the main idea of this Section. In the following, we reformulate the three rightmost terms one by one, making them more concrete such that we can compute them. The main outcomes are presented in the accompanying Tables; other used diagnosed values can readily be obtained from the available code (python notebook).

**B1    Effect 1: changing SST**

First, we estimate the term corresponding to effect 1, changing SST, at a fixed time of interest $t^*$ and depth 3.1 km by:

$$\Delta T_{\text{SST}}(t^*) \approx \sum_{i=1}^{8} \overline{\Delta T_i}(t^*) \cdot f_i(t_0),$$





where $f_i$ is the fraction of water mass $i$ (between 0 and 1) taken as a basin average at the 3.1 km depth layer and computed at steady state $t_0$. The average $\overline{\Delta T_i}(t^*)$ corresponds to the average surface temperature anomaly of dye tracer $i$ averaged over the
time interval $[t^* - a_{\max}, t^* - a_{\min}]$, where $a_{\min}$ and $a_{\max}$ are the minimum and maximum ideal age at 3.1 km depth occurring in the basin of interest. This is just one of the possible choices that can be made in order to estimate the history of surface temperatures $\overline{\Delta T_i}(t^*)$ that water mass $i$ has seen at the times of subduction. This choice corresponds to picking a certain time interval in Fig. 1b before the time step of interest $t^*$, which should represent the times of subduction. Here we chose a non-weighted average over all possible times of subduction based on the ideal age tracer $a$. Note that this is an approximation,
because the true age of, e.g., the NADW fraction of a certain water parcel is unknown. We only have the ideal age tracer at our disposal, which is the average over the entire probability density function of ages within a water parcel (grid cell) thus, e.g., also takes the age of the AABW fraction of this water parcel into account. A second assumption in our approach is taking the temperature of the surface where a water mass originates as the typical water mass temperature.

Writing out the averaging procedure,

$$\overline{\Delta T_i}(t^*) = \frac{1}{(t^* - a_{\min}) - (t^* - a_{\max})} \int\limits_{t^* - a_{\max}}^{t^* - a_{\min}} \Delta T_i(t)\, dt$$

and combining this with the previous gives one final Equation for effect 1:

$$\Delta T_{\mathrm{SST}}(t^*) \approx \frac{1}{a_{\max}(t_0) - a_{\min}(t_0)} \sum_{i=1}^{8} f_i(t_0) \int\limits_{t^* - a_{\max}(t_0)}^{t^* - a_{\min}(t_0)} \Delta T_i(t)\, dt \tag{B1}$$

Here, we emphasized that $a_{\min}$ and $a_{\max}$ are diagnosed at time $t_0$ for effect 1. If either of the times at the integral boundaries occurs before the start of the simulation $t_0$, then the values at $t_0$ are taken, which are representative since the ocean was in
steady state before $t_0$. The results of applying Eq. (B1) are shown in Table B1.

**B2    Effect 2a: changing water masses**

For effect 1, we multiplied the deviation in water mass temperature with the water mass fraction at steady state. For effect 2a, changing water masses, we now do the opposite: we take the product of the steady state water mass temperature with the deviation of the water mass fraction, again at a time of interest $t^*$ and at depth 3.1 km. This gives us the change in temperature
at 3.1 km due to water mass changes:

$$\Delta T_{\mathrm{WM}}(t^*) \approx \sum_{i=1}^{8} T_i(t_0) \cdot (f_i(t^*) - f_i(t_0)), \tag{B2}$$

where $T_i(t_0)$ is the typical temperature of water mass $i$ at steady state $t_0$, based again on its surface temperature. Note that $T_i(t_0)$ relates to $\Delta T_i(t)$ in Eq. (B1) as $\Delta T_i(t) := T_i(t) - T_i(t_0)$. The results are shown in Table B2. The choice of basin does not influence $T_i$ by definition, since the water mass temperature is diagnosed on its specific surface (Fig. A1).





**Table B1.** Computing effect 1, changing SST. The six columns in the middle present a subset of the diagnosed values needed for the computation: as an example, only water masses NADW and SO and only $t^* = 2000$ are shown. The two rightmost columns show the result of effect 1 by using Eq. (B1), where all 8 water masses were taken into account.

| Effect 1 | $a_{min}(t_0)^a$ | $a_{max}(t_0)^a$ | $\overline{\Delta T_{NADW}(2000)}^b$ | $\overline{\Delta T_{SO}(2000)}^b$ | $f_{NADW}(t_0)$ | $f_{SO}(t_0)$ | $\Delta T_{SST}(1750)^b$ | $\Delta T_{SST}(2000)^b$ |
|---|---|---|---|---|---|---|---|---|
| Atlantic | 13 | 746 | –0.0242 | –0.0419 | 0.580 | 0.307 | –0.0485 | –0.0333 |
| Pacific | 528 | 1397 | 0.00948 | –0.00662 | 0.149 | 0.780 | 0.000108 | –0.00462 |
| SO | 252 | 885 | –0.0535 | –0.0603 | 0.157 | 0.788 | –0.00972 | –0.0611 |

$^a$ Ages $a_{min}$ and $a_{max}$ are in years.
$^b$ Averaged temperatures $\overline{\Delta T_i}$ and results $\Delta T_{SST}$ are in $^\circ$C.

**Table B2.** Computing effect 2a, changing water masses. The six columns in the middle present a subset of the diagnosed values needed for the computation: as an example, only water masses NADW and SO are shown. In addition, $f_i(t_0)$ from Table B1 are used. The two rightmost columns show the result of effect 2a by using Eq. (B2), where all 8 water masses were taken into account.

| Effect 2a | $T_{NADW}(t_0)^a$ | $T_{SO}(t_0)^a$ | $f_{NADW}(1750)$ | $f_{SO}(1750)$ | $f_{NADW}(2000)$ | $f_{SO}(2000)$ | $\Delta T_{WM}(1750)^a$ | $\Delta T_{WM}(2000)^a$ |
|---|---|---|---|---|---|---|---|---|
| Atlantic | 5.486 | 2.732 | 0.584 | 0.301 | 0.568 | 0.319 | 0.0352 | –0.0203 |
| Pacific | 5.486 | 2.732 | 0.148 | 0.783 | 0.148 | 0.783 | –0.0113 | –0.0134 |
| SO | 5.486 | 2.732 | 0.154 | 0.793 | 0.159 | 0.786 | –0.0290 | 0.0222 |

$^a$ Temperatures are in $^\circ$C.

## B3 Effect 2b: changing water age

Changing ocean circulation consists of the combined effect of a) changing water masses (diagnosed by dye tracers) and b) changing water age (diagnosed by the ideal age tracer). If we adapt the simplified view of two deep water formation end members or pumps, located in the North Atlantic and in the Southern Ocean, then this distinction corresponds to a) changing the ocean volume filled by each pump (one of the pumps fills more of the ocean volume and the other less) and b) both pumps increase (decrease) equally in strength such that they turn over faster (slower) within their own overturning cell, which causes a younger (older) water age. It is in principle possible that one of these effects occurs independently without the other, but in reality they typically occur simultaneously and are hard to separate; when an overturning cell increases or decreases in strength,





it usually also becomes deeper or shallower. We try this separation nevertheless and estimate the effect of changing water age, representing absolute overturning rate.

Changing water age alters effect 1, changing SST. Since the ages $a_{\min}$ and $a_{\max}$ shift towards younger (older) values under stronger (weaker) absolute overturning, the average $\overline{\Delta T}(t_0)$ in Eq. (B1) must now be evaluated at $t^*$ instead of $t_0$. This gives the effect of water age

$$\Delta T_{\mathrm{WA}}\left(t^*\right) \approx \left(\frac{1}{a_{\max}(t^*) - a_{\min}(t^*)} \sum_{i=1}^{8} f_i(t_0) \int_{t^*-a_{\max}(t^*)}^{t^*-a_{\min}(t^*)} \Delta T_i(t)\, dt\right) - \Delta T_{\mathrm{SST}}\left(t^*\right), \tag{B3}$$

where we subtracted effect 1 in order to find only the adjustment due to effect 2b. Since the resulting contribution of water age
changes $\Delta T_{\mathrm{WA}}(t^*)$ is relatively small, the diagnosed values of $a_{\min}(t^*)$, $a_{\max}(t^*)$ and $\overline{\Delta T_i}(t^*)$ needed for the calculation are not shown here. The result $\Delta T_{\mathrm{WA}}(t^*)$ is displayed directly in summary Table B3.

**B4   Interaction**

The interaction is small, but computed for completeness by combining the deviations of $T_i$ and $f_i$ and summing again over all water masses $i$:

$\Delta T_{\mathrm{int}}\left(t^*\right) \approx \sum_{i=1}^{8} \overline{\Delta T_i}(t^*) \cdot \left(f_i(t^*) - f_i(t_0)\right)$           (B4)

All quantities required for this were already diagnosed and (partly) shown in Table B1 and Table B2. The result of Eq. (B4) is directly shown below in the respective column in Table B3.

**B5   Summarizing the results**

We summarize the above and compare the contribution of each effect in Table B3. Table 2 of the main text follows from Table
B3 by converting Celsius to centi-Kelvin, by rounding to 2 significant figures with a maximum of 3 decimals and by adding percentages. Finally, we make a few closing remarks about uncertainty.

    We note that the estimate of effect 1 (changing SST) has a large uncertainty, which explains a substantial amount in the residual column. This is due to the usage of the ideal age tracer as estimate for the age of a specific water mass, as mentioned above. For example: a water parcel with 75 % NADW with age 200 years and 25 % AABW with age 600 years gives an ideal
age of 300 years. In this example, effect 1 is estimated with an NADW age of 300 years and an AABW age of 300 years. This uncertainty of effect 1 can be quantified, because we can also compute effect 1 directly from the output of OcFIX, which gives the 'perfect' SST contribution, i.e., exactly as simulated in the Bern3D model. For completion, we list the values of the perfect SST contribution here, in the same order as the rows in Table B3: –0.00452, –0.0125, –0.0507, –0.0680, –0.017 and –0.0265 °C. Using these perfect values for the SST contribution would result in better residual values: –0.0100, –0.0136,
0.0040, 0.0018, –0.0163 and 0.0115 °C. We did not use these values for consistency reasons, as they do not correspond to the estimates for effect 2b, which is based on effect 1. We conclude that in most cases one third to two thirds of the residual is explained by the uncertainty in effect 1.





Effect 2a (changing water masses) is thought to be well-known, since it does not use the ideal age tracer to estimate water mass age. The uncertainty in effect 1 propagates in effect 2b (changing water ages), since it is based on effect 1. As an

uncertainty estimate for effect 2b, we take the percentage uncertainty in the SST contribution quantified above and take the same percentage as uncertainty for effect 2b. This gives an uncertainty of roughly 200 % to 300 % for most basins and most time steps. Note that this uncertainty is not symmetric, since the SST estimate is generally too small thus we also expect effect 2b to be too small rather than too large (in absolute values). Scaling the percentage contribution of effect 2b (Table 2) by a factor of 2 or 3 typically changes values of 2 % to maximally 6 % hence still very small. We conclude that the uncertainty in

effect 2b is large, but this does not change our main message of effect 2b being small in all cases, except perhaps in the Atlantic in 2000 CE.

**Table B3.** Summary of quantification of how different effects act on the deep ocean temperature at 3.1 km, per basin and at times of interest 1750 CE and 2000 CE. All values are in °C.

| **Multiple effects @3.1 km on:** | | **1. Changing SST**[a] 1. History of SST | **2. Changing circulation**[b] 2a. Water masses[c] | 2b. Water age[d] | **Interaction**[e] | **Residual**[f] | **Total**[g] |
|---|---|---|---|---|---|---|---|
| Atlantic: | 1750 | –0.0485 | 0.0352 | –0.00168 | –0.00004 | 0.00178 | –0.0133 |
| | 2000 | –0.0333 | –0.0203 | –0.0498 | –0.00025 | –0.03687 | –0.1405 |
| Pacific: | 1750 | 0.000108 | –0.0113 | 0.000233 | –0.000001 | –0.01462 | –0.0256 |
| | 2000 | –0.00462 | –0.0134 | –0.000375 | –0.00002 | –0.02152 | –0.0400 |
| Southern Ocean: | 1750 | –0.00972 | –0.0290 | –0.000587 | –0.00006 | –0.02541 | –0.0648 |
| | 2000 | –0.0611 | 0.0222 | 0.0000831 | –0.00006 | 0.04606 | 0.00724 |

[a] Results are taken over from Table B1.

[b] Changing circulation consists of the – not necessarily independent – subeffects 2a and 2b.

[c] Results are taken over from Table B2.

[d] The water-age contribution follows from Eq. (B3).

[e] The interaction is the cross term of deviations in changing SST and in changing water masses and follows from Eq. (B4).

[f] The residual equals the total minus effects 1, 2a, 2b and their interaction. It represents unaccounted effects and uncertainties in the calculations.

[g] The total of all effects is given by the model output $\Delta T := T(t^*) - T(t_0)$ of OcTRA, which is averaged at 3.1 km within the basin of interest.



*Author contributions.* J.S. carried out the numerical simulations, implementation and analysis. T.F.S. conceived the original idea and supervised the project. J.S. lead the writing of the manuscript. Both authors contributed to the interpretation of the results.

*Competing interests.* The authors declare that they have no conflict of interest.

*Acknowledgements.* This work is supported by the Swiss National Science Foundation by grant SNF 200020_172745. We gratefully acknowledge Thomas L. Frölicher for helpful comments on an earlier version of the manuscript.





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
