# Peer review of "Effect of changing ocean circulation on deep ocean temperature in the last millennium"

_Earth System Dynamics, 2020_

## Referee Comment (RC1) · Anonymous Referee #1 · 1 Jul 2020

The authors compare the contribution of surface temperature and of the ocean circulation changes in response to a moderate radiative perturbation to temperature variations in the deep ocean. This study is welcome as it is often assumed that, in case of a relatively small changes in circulation, the effect of those changes can be neglected and we can consider that the signal at depth can be simply interpreted as the result of the transport of anomalies created at surface by the mean circulation. By doing some well-designed experiments and precise diagnostics, the authors show that this approximation of constant circulation leads to an underestimation of the changes and biases in the timing of warming and cooling at many locations. The manuscript is well-written and provides the adequate information. I have only a few suggestions below on the

structure of the paper or to improve the clarity of some parts but none is mandatory for me before publication.

General comment

1/ I think Table 2 is hard to follow without reading the sup. mat. Furthermore, the uncertainties are very large (as shown by the residuals) and it does not bring much new information. The diagnostic related to water age could be instructive but, except in one case, the value is very small and very uncertain (see lines 306-307 and line 536). I thus suggest to move it to sup. mat. So, all the information related to those diagnostics is grouped at one location. The results could be mentioned in section 3.4 but without details. It also implies that the values of effect one can be computed as given line 528 and display a lower uncertainty.

Specific comments

1/ Line 25. I guess the sentence refers to the ocean depth at the core location but the majority of the cores record changes in SST or near surface temperature changes.

2/ Line 107. I am not sure I understand what is meant by 'at every surface grid cell are set to a latitude-longitude field'. Does it mean that a constant spatial field is used and the value is different for each grid cell? Please rephrase.

3/ Line 115. Maybe I missed something in the procedure but is it possible that a drift occurs just because of the shift to fixed ocean velocities in OcFIX. Is this taken into account?

4/ Line 134. The comparison of results of this study with the ones of Collins et al. is made for different conditions and time periods. Would it be possible to compare the results presented here for the 20th century with the trends for the 20th century simulated in other models for a more meaningful comparison ?

5/ Line 141. Isn't it Fig 2a ?

6/ Line 166. The link with Fig A3 is not straightforward as you did not explain that it corresponds to tracer concentration while the reader could have imagine that it relates directly to 'persistent cold anomaly'.

7/ Line 171. Maybe it would be useful to discuss the ventilation rate in the Southern Ocean in the model and its realism as it may have an impact on the results.

8/ Line 175. What is the 'TRA minimum' ?

9/ Line 212. Is it the southward heat transport?

10/ Line 219. Why is there a difference of one order of magnitude in the results while the difference in the forcing is only a factor 3?

11/ Why is it surprising ?

12/ Line 356. 600 CE is usually not associated with a medieval warm period. The classical 'Medieval Climate Anomaly' is rather around 850-1150 CE.

13/ Line 357 and below (e.g. 364-365). The results of GH19 seems to be well in agreement with the observations. I think this should be mentioned more explicitly and discussed.

14/ Line 369-370. This model bias provides very important information. For me, this has to be explained earlier in order to analyze the implications for all the diagnostics presented.

15/ Figure 11. It is not easy to compare with the Figure 2 of GH2019. Could it be possible to reproduce for instance on figure 11 the anomalies deduced from observations (or the results of GH2019 if publicly available ) ?

16/ Line 377. Is it 'already warming' or 'warmer conditions than in 1875' ?

17/ Line 385. As the model is fast, would it be possible to make an additional experiment to prove this hypothesis ?

[Figure]

18/ Line 413. For me, with the experimental design, SSTs are the same but the heat uptake can be different (as circulation for instance can be different).

---

## Referee Comment (RC2) · Anonymous Referee #2 · 3 Aug 2020

This paper analyzes the effect of ocean circulation changes on the transfer of surface temperature anomalies to the deep ocean on time scales of centuries to millennia. It is motivated by an earlier study of Gebbie and Huybers (GH19, see references), which was based on a model with constant ocean circulation as inferred from modern observations by inversion. This is an important question for understanding deep ocean climate evolution, but has also consequences beyond that, for example, how important is this history for the initialization of "historical" simulations.

Scheen and Stocker apply first a coupled model of intermediate complexity under idealized radiative forcing to derive surface conditions that mimic the transition to and from

the Little Ice Age towards modern global warming and beyond. Then they run ocean-only experiments driven by the surface conditions from the coupled model discriminating between simulations with fixed circulation and free currents. The experimental set-up is well designed and illustrates the long-term behavior of temperature trends at depth in the deep ocean. They find that circulation changes do play a role and may lead to quite different results from GH19. The authors provide some sensitivity studies to explore parameter uncertainty. Thus the tools are quite adequate and they are used in a clever and novel way.

The manuscript is written concisely and brings the message across. The abstract provides a good summary. The material and figures are illustrative to support the hypothesis. I find the paper quite convincing but suggest that the authors discuss a bit more the possible shortcomings of their approach.

In conclusion, I recommend to accept the manuscript after minor revisions.

General comments:

The authors describe that the coupled model is run to equilibrium and then the ocean only model is run "using steady state SST and SSS". Are there other drivers of the ocean model, e.g. momentum flux from the coupled model, or are winds and /or other fluxes prescribed externally?

If I understand correctly, the authors can in this set-up only study the effect of density changes on the circulation and the resulting feedback on temperature and salinity transfers. But in the "real" world of the last millennium, for example strong volcanic eruptions may have changed the dynamics of ocean and atmosphere for decades to centuries. It should be mentioned that such effects are not included here.

How different are the circulation and simulated features in the coupled simulation, compared to the ocean stand-alone run? Is the AMOC or meridional heat transport history similar? A large part of the finding depends on the history of the AMOC in the model.

The AMOC slightly strengthens during the LIA cooling and then weakens under modern warming. Does this mean that the cooling/warming drives the AMOC in this model? One could also assume that the cold surface conditions in the LIA North Atlantic were a result of weak AMOC and slow heat transports as has been put forward by many authors. Would the findings of this study still hold?

It is said that "simulations with fixed ocean circulation are implemented by using diagnosed values at every grid-point from the steady state". Which variables are used, only horizontal velocities, or also diffusion/mixing coefficients?

Minor issues:

Ln 73: :idealized forcing: was the forcing tuned in this way to get the desired SST time series to be similar to GH19 or is it based on estimates how the radiative forcing actually changed?

Ln 107: "at every grid cell are set to a latitude-longitude field": this is very unclear.

Ln 219: Was the Winton experiment a 1% $CO_2$ increase per year experiment?

Ln 221: should be: "column integrated heat content per unit area"

---

## Author Comment (AC1) · 11 Sep 2020

**Reply to review comments**

We thank both reviewers for their time and for their constructive comments, which helped to improve the formulation and content of the manuscript.

Listed below are: the reviewers' comments in black, our reply in blue, "quotes from the originally submitted manuscript in brown" and "quotes from the revised manuscript in green". All line numbers follow the numbering in the originally submitted manuscript. Figures in the paper are referred to as Fig. 1, Fig. 2, etc. and Figures in this reply as Fig. R1, Fig. R2, etc.

**Anonymous Reviewer #1**

The authors compare the contribution of surface temperature and of the ocean circulation changes in response to a moderate radiative perturbation to temperature variations in the deep ocean. This study is welcome as it is often assumed that, in case of a relatively small changes in circulation, the effect of those changes can be neglected and we can consider that the signal at depth can be simply interpreted as the result of the transport of anomalies created at surface by the mean circulation. By doing some well-designed experiments and precise diagnostics, the authors show that this approximation of constant circulation leads to an underestimation of the changes and biases in the timing of warming and cooling at many locations. The manuscript is well-written and provides the adequate information. I have only a few suggestions below on the structure of the paper or to improve the clarity of some parts but none is mandatory for me before publication.

We thank the reviewer for acknowledging the impact of our manuscript and for the detailed feedback, which improved the clarity and correctness of the text.

**General comment**

1. I think Table 2 is hard to follow without reading the sup. mat. Furthermore, the uncertainties are very large (as shown by the residuals) and it does not bring much new information. The diagnostic related to water age could be instructive but, except in one case, the value is very small and very uncertain (see lines 306-307 and line 536). I thus suggest to move it to sup. mat. So, all the information related to those diagnostics is grouped at one location. The results could be mentioned in section 3.4 but without details. It also implies that the values of effect one can be computed as given line 528 and display a lower uncertainty.

   Table 2 and most accompanying text is moved to the supplement in a new subsection B5 (Table 2 is merged here with Table B3). This shortens Section 3.4. We did not change the values of effect 1 in the table, because that would make it inconsistent, i.e., rows not adding up to 100%. We think the large uncertainty is discussed clearly enough.

**Specific comments**

1. Line 25. I guess the sentence refers to the ocean depth at the core location but the majority of the cores record changes in SST or near surface temperature changes.

   We changed L24-26 to:

   "(...), but oceanic reconstructions below the sea surface are scarce for the last millennium (Moffa-Sánchez et al., 2019)."

2. Line 107. I am not sure I understand what is meant by 'at every surface grid cell are set to a latitude-longitude field'. Does it mean that a constant spatial field is used and the value is different

for each grid cell? Please rephrase.

For every time step we use a new spatial field. We clarified the formulation to:
"[the SST, SSS and sea ice] of surface grid cells are set to the corresponding value in the (latitude, longitude, time step) boundary conditions that are obtained from (...)"

3. Line 115. Maybe I missed something in the procedure but is it possible that a drift occurs just because of the shift to fixed ocean velocities in OcFIX. Is this taken into account?

Switching to FIX mode cannot be the cause of the drift, since we also observed a drift in TRA simulations (before doing the extended spin-up). During fixed-circulation runs, the circulation of every time step of the first simulation year is saved and repeated automatically in subsequent years. Thus the first simulation year should be without forcing, which is ensured by our zero forcing from 1200 to 1223 CE. Small differences in AMOC between simulation year 1 and all subsequent years can occur around the $4^{th}$ digit, but these are not visible by eye in, e.g., Fig. 2.

4. Line 134. The comparison of results of this study with the ones of Collins et al. is made for different conditions and time periods. Would it be possible to compare the results presented here for the 20th century with the trends for the 20th century simulated in other models for a more meaningful comparison ?

"[about 1750-2000 CE:] This 1.5 Sv AMOC slowdown is within the CMIP5 range of fully coupled AOGCMs and the AMOC in that model mean experiences a similar decrease of 2 Sv from 2000 to 2100 CE under low emission forcing (Collins et al., 2019)."

This is a good point. We now omitted the second part of this sentence. We were mainly interested to compare 1750-2000 CE, because the AMOC slows down in this period in our set-up, but CMIP5 models give a broad range here.

5. Line 141. Isn't it Fig 2a ?

Correct, done. Thank you.

6. Line 166. The link with Fig A3 is not straightforward as you did not explain that it corresponds to tracer concentration while the reader could have imagine that it relates directly to 'persistent cold anomaly'.

We clarified this.

7. Line 171. Maybe it would be useful to discuss the ventilation rate in the Southern Ocean in the model and its realism as it may have an impact on the results.

We added the following here:
"This ventilation rate in the SO (Fig. 8d and 9d) agrees well with radiocarbon-inferred observations (Gebbie and Huybers, 2012) in the average and maximum SO water age, although qualitatively the Bern3D deep water formation occurs too far south, since it is enhanced with a prescribed salt flux in the Ross and Weddell Seas."

8. Line 175. What is the 'TRA minimum' ?

We changed "TRA minimum" to "temperature minimum in TRA".

9. Line 212. Is it the southward heat transport?

Yes, we replaced "southward transport" with "southward heat transport".

10. Line 219. Why is there a difference of one order of magnitude in the results while the difference in the forcing is only a factor 3?

We now added:
"The redistribution heat transport could also be smaller in our case because atmospheric processes are better represented in GCMs (general circulation models), such as the GFDL model used in Winton et al. (2013). The hydrological cycle of the Bern3D EMIC model reacts too weakly compared to another GCM, CESM version 1.0.1, for an ensemble member over the past millennium. Here the Bern3D SSS anomalies were an order of magnitude too low, although showing the same qualitative patterns."

11. Why is it surprising ?

This could refer either to "surprising" in line 288 or to "surprisingly" in line 309. Line 288 addresses 2000 CE:
"It is surprising that in the Pacific (excluding the SO sector), the same qualitative water mass

anomalies as in 1750 are still present, however displaced northwards, and opposite (...)"
This is different than what we saw in the Atlantic and SO, but may not be surprising indeed. We reformulated this to:
"In the Pacific (excluding the SO sector), the same qualitative water mass anomalies as in 1750 are still present, however displaced northwards. This slower water mass renewal is due to the longer residence time of waters in the Pacific. Opposite (...)"
Line 309 (which is now moved to the appendix; see general comment #1):
"Surprisingly, the change in absolute overturning rate, which is estimated via water age, has practically no influence in all cases except the Atlantic in 2000 CE."
is now changed to:
"The change in absolute overturning rate, which is estimated via water age, has practically no influence in all cases except the Atlantic in 2000 CE, where the largest overturning anomalies take place."

12. Line 356. 600 CE is usually not associated with a medieval warm period. The classical 'Medieval Climate Anomaly' is rather around 850-1150 CE.
We agree that the MCA or medieval warm period is usually defined later than 600 CE, but we followed Gebbie and Huybers (2019) (GH19 hereafter) such that we can compare our results with theirs. Figure 1a of GH19 shows a warm period (i.e. global average warmer than 0 CE) from ca. 0 to 1200 CE with the maximum temperature in the middle, at 600 CE. Note that this line is now reformulated due to specific comment #17.

13. Line 357 and below (e.g. 364-365). The results of GH19 seems to be well in agreement with the observations. I think this should be mentioned more explicitly and discussed.
We made a few changes around these lines and emphasized that the model of GH19 corresponds well with measured water ages. We have the opinion though that both models have their strengths and weaknesses (static circulation in model of GH19 as measured during transient (early) industrial warming; less realistic circulation/ventilation rates in Bern3D) and none of them represents the truth.

14. Line 369-370. This model bias provides very important information. For me, this has to be explained earlier in order to analyze the implications for all the diagnostics presented.
"(...) because the deep North Atlantic is too old in the Bern3D ocean model compared to radiocarbon observations (Gebbie and Huybers, 2012) (...) These observations show radiocarbon ages of up to 600 years in the North Atlantic (zonal average), whereas we find ideal ages over 900 years (Fig. 8)."
We analyzed the impact of this model bias in more detail and found that it has a smaller impact than we expected. We mention this now briefly in the discussion instead and give more details here: We considered a sensitivity simulation with a smaller bias in North Atlantic water age to see if the deep Atlantic would then be already warming at present, as in GH19. Namely, the maximum age of Atlantic water in OcFIX_strongmix is ca. 800 yr (Fig. R1d) instead of 1000 yr for STD. Although still too old, this compares better to the observations. Since the downward temperature propagation for strongmix (Fig. R2) is very comparable to STD (Fig. 4a, b), we conclude that this model bias is likely not responsible for the differences in downward propagation seen between GH19 and our study.

15. Figure 11. It is not easy to compare with the Figure 2 of GH2019. Could it be possible to reproduce for instance on figure 11 the anomalies deduced from observations (or the results of GH2019 if publicly available ) ?
We agree with the reviewer and added a third panel to Fig. 11 with the model results of GH19, Fig. 2. We thank G. Gebbie for providing the data and details on the analysis.

16. Line 377. Is it 'already warming' or 'warmer conditions than in 1875' ?
'Warmer conditions than in 1875' is indeed more correct. We clarified in lines 374-379 that Fig. 11 only shows warming/cooling trends between 1875 and 1995 CE.

17. Line 385. As the model is fast, would it be possible to make an additional experiment to prove this hypothesis ?
We did so and rejected this hypothesis. We now mention in the text instead that including a medieval warm period (MWP) does not alter our main results and provide Fig. R3 and Fig. A6. We performed a coupled simulation OcAtmTRALong starting at 0 CE with a MWP. For consistency

with the later simple evolution, the MWP radiative forcing from 0 to 1200 CE is chosen triangular-shaped with a maximum of 0.35 $Wm^{-2}$ at 600 CE. From 1200 CE the forcing is identical to the forcing in Fig. 1a. OcAtmTRALong supplied the SST, SSS and sea ice boundary conditions for the ocean-only simulations with MWP: OcTRALong and OcFIXLong. We reproduced Fig. 11 now for these runs in Fig. R3. The cooling in the North Pacific in Fig. R3 did not become as strong as in GH19 (i.e., not crossing the –5 cK contour line), which disproves the hypothesis in L385.
For completion we also reproduce Fig. 4a, b for OcTRALong and OcFIXLong in Fig. A6 (note the different time axis). These figures are very similar from ca. 1600 CE onward with the notable difference that the cooling in the Pacific does not reach below 2 km anymore for OcFIXLong (in the case of OcFIX the cooling here was less than 5 cK).

18. Line 413. For me, with the experimental design, SSTs are the same but the heat uptake can be different (as circulation for instance can be different).
"In this paper, we held ocean-atmosphere feedbacks and total ocean heat uptake identical between transient and fixed simulations, by prescribing the same SST and SSS time series."
That is correct and visible in Fig. 7a, which shows different OHC anomalies for OcTRA (total; black line) and OcFIX (changing SST; blue part). We do not mention total ocean heat uptake/content anymore in L413 and L419. We also corrected L43 from "we cannot quantify differences in global ocean heat uptake" to "we cannot quantify differences in global ocean heat uptake that arise from different ocean-atmosphere interactions" . In Fig. 6 and 7 we do actually quantify OHC changes, which in our case differ between TRA and FIX only because of ocean circulation changes. Our ocean-only setup is as if the atmosphere would be an infinite source of heat/cold that provides the heat/cold needed to overwrite the temperature in the surface layer to the desired imposed SST. The amount needed differs between OcTRA and OcFIX because of circulation changes, which determine how much heat/cold is transported down from the surface layer. Evidently, in OcTRA more heat/cold enters the ocean than in OcFIX (see, e.g., Fig. 7a and the larger TRA amplitudes in Fig. 4d).

**Anonymous Reviewer #2**

This paper analyzes the effect of ocean circulation changes on the transfer of surface temperature anomalies to the deep ocean on time scales of centuries to millennia. It is motivated by an earlier study of Gebbie and Huybers [Gebbie and Huybers (2019)], which was based on a model with constant ocean circulation as inferred from modern observations by inversion. This is an important question for understanding deep ocean climate evolution, but has also consequences beyond that, for example, how important is this history for the initialization of "historical" simulations. Scheen and Stocker apply first a coupled model of intermediate complexity under idealized radiative forcing to derive surface conditions that mimic the transition to and from the Little Ice Age towards modern global warming and beyond. Then they run ocean-only experiments driven by the surface conditions from the coupled model discriminating between simulations with fixed circulation and free currents. The experimental set-up is well designed and illustrates the long-term behavior of temperature trends at depth in the deep ocean. They find that circulation changes do play a role and may lead to quite different results from GH19. The authors provide some sensitivity studies to explore parameter uncertainty. Thus the tools are quite adequate and they are used in a clever and novel way. The manuscript is written concisely and brings the message across. The abstract provides a good summary. The material and figures are illustrative to support the hypothesis. I find the paper quite convincing but suggest that the authors discuss a bit more the possible shortcomings of their approach. In conclusion, I recommend to accept the manuscript after minor revisions.
We thank the reviewer for this positive evaluation and for the concise summary. We now mention the implication for the initialization of historical simulations in the text. In addition to the points addressed below, we now also discuss another shortcoming in more detail, see #10 of reviewer #1.

**General comments**

1. The authors describe that the coupled model is run to equilibrium and then the ocean only model is run "using steady state SST and SSS". Are there other drivers of the ocean model, e.g. momentum flux from the coupled model, or are winds and /or other fluxes prescribed externally?

The Bern3D model is forced with a seasonally varying wind field climatology at the sea surface, which is unchanged throughout simulations. The wind field is identical for coupled and ocean-only simulations. We now added this in the text.

There are no other drivers, except for sea ice, which was already mentioned in the text. In the coupled model, sea ice is dynamically simulated where the atmospheric temperature is cold enough. Since diagnosing atmospheric temperature is not possible for ocean-only, a time-varying sea ice area was implemented as boundary condition (from OcAtmTRA; together with SST and SSS).

2. If I understand correctly, the authors can in this set-up only study the effect of density changes on the circulation and the resulting feedback on temperature and salinity transfers. But in the "real" world of the last millennium, for example strong volcanic eruptions may have changed the dynamics of ocean and atmosphere for decades to centuries. It should be mentioned that such effects are not included here.

Long-term effects of volcanic forcing are implicitly included in the LIA segment of the total radiative forcing we use. Pulse-like volcanic coolings are not considered, since we excluded decadal oscillations on purpose in our idealized forcing in order to obtain a clean and less complex signal of downward temperature propagation of longer cold and warm spells. We already mention in L77:

"Possible radiative forcings due to land-use change, volcanic eruptions or solar irradiation are not considered in our illustrative scenario."

3. How different are the circulation and simulated features in the coupled simulation, compared to the ocean stand-alone run? Is the AMOC or meridional heat transport history similar? A large part of the finding depends on the history of the AMOC in the model.

The evolution of MOC during the coupled runs is given in Fig. R4. These time series are very similar to the ocean-only situation (Fig. 2) except that they start at different steady state values. The difference between the coupled and ocean-only situation in steady state is as described in line 115-119:

"During the extended spin-up, adjustments of the global ocean circulation take place: the steady state Atlantic meridional overturning circulation (AMOC) maximum decreases by 1.5 Sv and the strength of the global meridional overturning circulation (MOC) decreases by 0.8 Sv. Deep ocean temperatures are 0.3 to 0.6°C colder below 3 km, because of a relative increase of Antarctic Bottom Water (AABW) with respect to North Atlantic Deep Water (NADW), and are up to 0.2°C warmer in the depth range from 500 to 3000 m due to an absolute decrease in NADW."

4. The AMOC slightly strengthens during the LIA cooling and then weakens under modern warming. Does this mean that the cooling/warming drives the AMOC in this model? One could also assume that the cold surface conditions in the LIA North Atlantic were a result of weak AMOC and slow heat transports as has been put forward by many authors. Would the findings of this study still hold?

The cooling/warming seems to be mainly driving the AMOC in our simulations, but it goes both ways since we also observed a warming hole in the North Atlantic SST caused by the AMOC response (not shown). If the AMOC would slow down more and drive more SST changes, possibly the results would be more like Winton et al. (2013), who find an even stronger redistribution heat transport (although other differences between the set-ups and GFDL vs. Bern3D exist).

5. It is said that "simulations with fixed ocean circulation are implemented by using diagnosed values at every grid-point from the steady state". Which variables are used, only horizontal velocities, or also diffusion/mixing coefficients?

Only the density is saved for every time step during 1 simulation year. The velocities in x-, y- and z-direction follow from frictional geostrophic and hydrostatic balance, which in turn determine the transport of temperature and salinity by advection, diffusion and convection. We added this briefly in the text.

All 3 diffusion/mixing coefficients of the Bern3D ocean (isopycnal, diapycnal and Gent-McWilliams diffusivity) are globally uniform and stay constant throughout the simulations for both TRA and FIX. Note that diapycnal diffusivity was varied in the sensitivity studies.

**Minor issues**

1. Ln 73: :idealized forcing: was the forcing tuned in this way to get the desired SST time series to be similar to GH19 or is it based on estimates how the radiative forcing actually changed?

We added:
"The forcing was chosen to get a global average SST time series that is comparable to Gebbie and Huybers (2019)."

2. Ln 107: "at every grid cell are set to a latitude-longitude field": this is very unclear.
   Reformulated, see specific comment #2 of reviewer 1.

3. Ln 219: Was the Winton experiment a 1% CO2 increase per year experiment?
   Yes. This information is now added in the text.

4. Ln 221: should be: "column integrated heat content per unit area"
   Done, thank you. We also changed the synonym "ocean heat density (OHD)" to "ocean heat areal density (OHD)".

[Figure]

Figure 1: *For specific comment #14 of reviewer #1:* reproduction of Fig. 8 (see caption there), now for runs OcTRA_strongmix and OcFIX_strongmix.

[Figure]

Figure 2: *For specific comment #14 of reviewer #1:* reproduction of Fig. 4a, b (see caption there), now for runs OcTRA_strongmix and OcFIX_strongmix.

[Figure]

Figure 3: *For specific comment #17 of reviewer #1:* reproduction of Fig. 11 (see caption there), now for runs OcTRALong and OcFIXLong, which include a medieval warm period between 0 and 1200 CE and from 1200 to 2800 CE have forcing identical to STD.

[Figure]

Figure 4: *For general comment #3 of reviewer #2:* reproduction of Fig. 2 (see caption there), now for the coupled runs OcAtmTRA and OcAtmFIX.

[revised manuscript text omitted]